# Clamping force control of electro–mechanical brakes based on driver intentions

Jing Li[1], Tong Wu[1]*, Tianxin Fan[1], Yan He[1], Lingshuai Meng[1], Zuoyue Han[2]

1 College of Automotive Engineering, Jilin University, Changchun, Jilin, China, 2 UISEE (Shanghai) Automotive Technologies Ltd, Shanghai, China

☯ These authors contributed equally to this work.

* wut18@mails.jlu.edu.cn

**Data Availability Statement:** All relevant data are within the manuscript and its Supporting Information files.

**Funding:** JL received support from the National Key R&D Program of China (2018YFB0105900).

## Abstract

Electro–mechanical brakes (EMBs) are the future of braking systems, particularly in commercial vehicles. Therefore, it is important to design a simple EMB scheme and establish its clamping force control strategy to satisfy the demands of commercial vehicle braking systems. This study proposes a pneumatic disc–brake–based EMB for an electric bus. Its working principle was established, and the system model was analyzed. Subsequently, the hidden Markov models (HMMs) of driver decelerate and brake intentions were built and recognized based on the analytic hierarchy process (AHP). Given the time–consuming behavior of the proposed EMB to eliminate brake clearance due to the leverage effect of the arm and motor performance limitation, a clamping force control strategy factoring in the driver intentions was developed to improve the response performance without changing the structure or size of the EMB. Furthermore, simulation analyses were performed using MATLAB/Simulink. The results confirmed that under the action of a step and 5 Hz triangular sawtooth signals, the clamping force output from the EMB corresponds well with the target signal. The clamping force gradually increases when approaching the target without overshoot and jitter during the process. The overall clamping force response time is decreased by approximately 0.25 s under the driver emergency brake than the conventional control method. Hence, the response performance of the EMB is improved.

## 1. Introduction

With rapid economic and social development, there is an ever–growing need for energy resources and safety [1, 2], particularly in the transportation sector [3]. The conventional hydraulic or pneumatic brake can–not address modern vehicle requirements such as precise control, rapid response, and high integration, particularly in electric and intelligent vehicles that require active safety and handing stability. Thus, further development of conventional braking systems is restricted, and brake–by–wire (BBW) systems have gained increased attention in the industry. Based on their structure, the BBW system can be divided into two: electro–mechanical brakes (EMBs) and electro–hydraulic brakes (EHBs). The EMB hinders brake pipelines and disconnects the mechanical connection between the brake pedals and actuators

The funders had no role in study design, data collection and analysis, decision to publish, or preparation of the manuscript. The UISEE (Shanghai) Automotive Technologies Ltd provided support in the form of salaries for author ZH, but did not have any additional role in the study design, data collection and analysis, decision to publish, or preparation of the manuscript. The specific roles of these authors are articulated in the 'author contributions' section.

**Competing interests:** The authors have declared that no competing interests exist. Author ZH was employed by UISEE (Shanghai) Automotive Technologies Ltd, UISEE (Shanghai) Automotive Technologies Ltd provided support in the form of salaries for author ZH, but did not have any additional role in the study design, data collection and analysis, decision to publish, or preparation of the manuscript. This does not alter our adherence to PLOS ONE policies on sharing data and materials.

completely. When the driver hits the brake pedal, the actuator operation at each wheel is controlled by an electric signal. Contrarily, the EHB hinders the vacuum booster, decouples the master cylinder from the brake pedal, and reserves the hydraulic pipelines to guarantee system reliability.

Upon further developments of the braking system, the EMB demonstrates a simple structure, flexible control, and convenience for integrating the electronic parking braking, anti-lock braking system, electronic stability control, traction control system, autonomous emergency braking, and autonomous driving system. The goal of EMB is to build an intelligent and integrated control platform of chassis, ensure active safety and handling stability, and improve the vehicle's overall performance.

Several enterprises, institutions, and scholars have been investigating the structural schemes of EMB. In general, a conventional EMB system is mainly composed of a power source, force amplification mechanism, motion conversion mechanism, brake disc, and caliper. Motors are used as the power source in most schemes, while a screw or bevel gear [6, 11] is used for the motion conversion mechanism. Particularly, the existing EMBs can be divided into two conventional schemes based on the force amplifying mechanism as shown in Fig 1. In scheme A, as shown in Fig 1a, a gear [4–7] or worm [8, 9] is used as the force amplifying mechanism; scheme B, shown in Fig 1b, utilizes the self–increasing effect of wedge mechanisms to increase clamping force [9–12]. During the operation of the system, the torque output from the power source is amplified by the force amplifying mechanism, converted into thrust force by the motion conversion mechanism, and acts on the brake shoes inside the caliper to generate the clamping force. Contrarily, scheme A is more direct in power transmission and simple in control, but the output capacity of the system is dependent on the motor and reduction ratio, which requires a higher performance of the motor. Although scheme B utilizes the wedge mechanism to increase the force, which demonstrates a flexible arrangement and lower requirements of the motor, the strong nonlinear characteristics of the wedge mechanism increases the control difficulty and decreases the braking stability [12]. For a new structure, Yu *et al.* designed an EMB based on the magnetorheological principle (MR) and wedge mechanism and analyzed its self–energizing and energy recovery effects [13].

The clamping force is the primary function that helps to understand various braking functions of the EMB system. Hence, the clamping force control has been extensively studied throughout this decade. In general, studies on the clamping force of the EMB primarily focus on force control based on sensors [4, 6, 14–21] and force estimation without sensors [10, 22–24]. Considering the studies on clamping force control, Hoseinnezhad *et al.* proposed a real-time measurement method, and calibrated an EMB characteristic curve based on the method [14]. Qiao, Jo *et al.* studied the influence of friction torque on the dynamic performance of the EMB, deduced a friction model, and presented a clamping force estimation and control method considering the friction of an actuator [4, 15]. Line *et al.* improved the clamping force control architecture and built a control strategy based on the model predictive control algorithm [16]. Krishnamurthy, Eum *et al.* proposed a robust control method and conducted related experiments [6, 17]. Atia *et al.* designed a sliding mode controller [18]. Kim *et al.* developed a control strategy based on the brain limbic system–genetic algorithm and compared it with the cascade control method [19]. Ahmad *et al.* established a floating caliper EMB model for the wedge mechanism and proposed a corresponding control method [20]. Park, Choi *et al.* designed a new clamping force controller to overcome the limitation of the response delay of existing solutions [21].

Considering the studies on clamping force estimation, Kwon, Lee *et al.* established a new EMB model, which divided clamping force into linear and nonlinear parts and used the Kalman filter algorithm to design a state estimator for evaluating the state of clamping force [10].

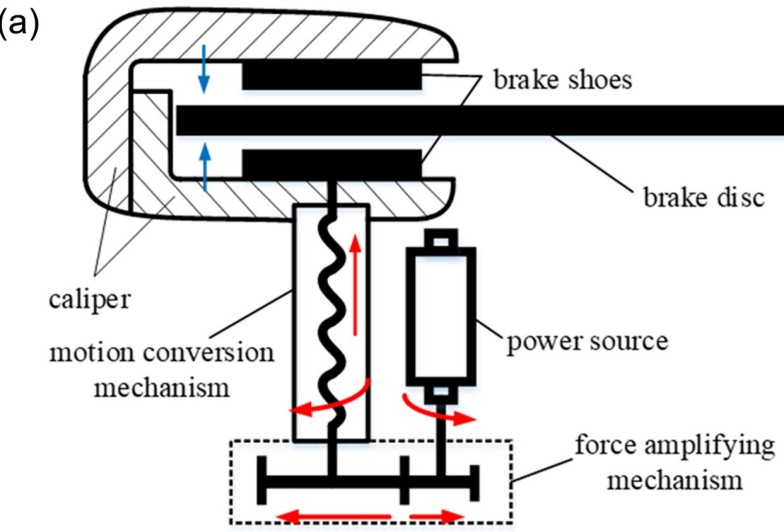

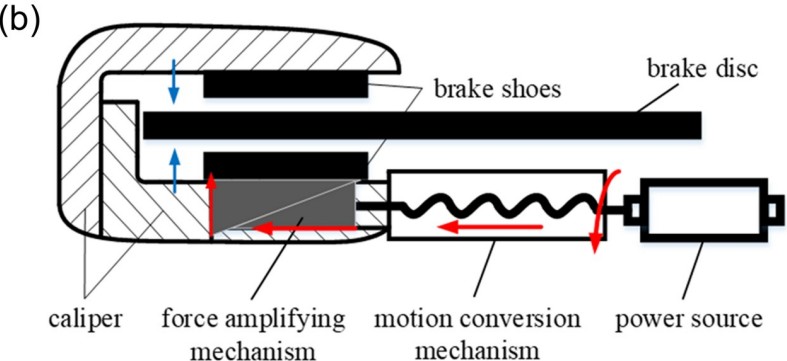

**Fig 1. Existing EMB schemes.** (a) Scheme A. (b) Scheme B.

Park *et al*. proposed a sensor–less estimation and control method for the self–energizing brake system [22]. Ki *et al*. built an EMB system model and provided an estimation method based on the hysteresis model to detect the contact point between the brake shoes and disc and estimate without sensors [23, 24].

Most studies on the clamping force control mainly focus on improving the system response and follow–up performance under typical signals. Although the clamping force estimation method can eliminate sensors, reduce costs, and further simplify the structure, the accuracy of estimation has a significant impact on the control effect. Thus, the reliability of this method must be verified.

Constraints in storing and sealing of hydraulic oil make it difficult to employ EHB in commercial vehicles, particularly in heavy commercial vehicles. Currently, most commercial vehicles are equipped with pneumatic brake systems. Compared with the pneumatic brake, the EMB does not require large and complicated pneumatic pipelines and components. Furthermore, the EMB could effectively reduce the body weight and complexity of the braking system, eliminate the exhaust noise during braking, and realize more accurate control and distribution of the braking force. Therefore, the EMB demonstrates broader prospects than the pneumatic brake in the applications of commercial vehicles. Currently, limited by the motor performance,

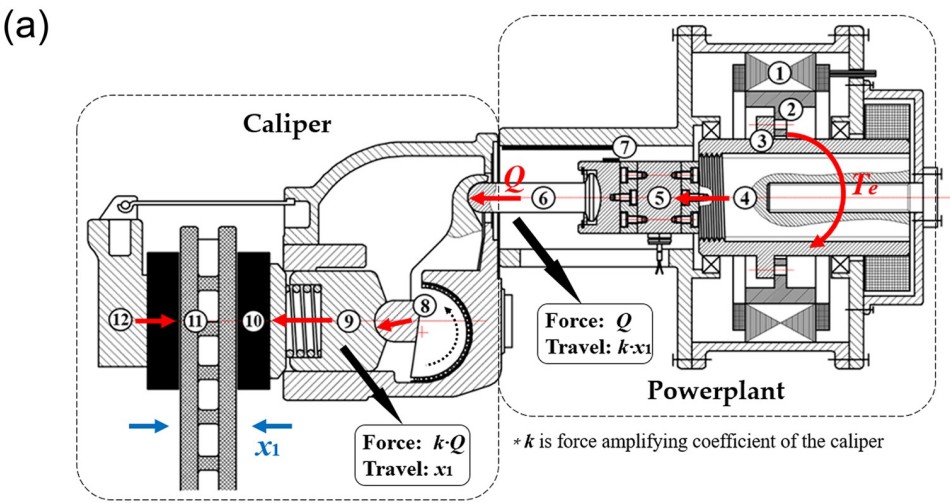

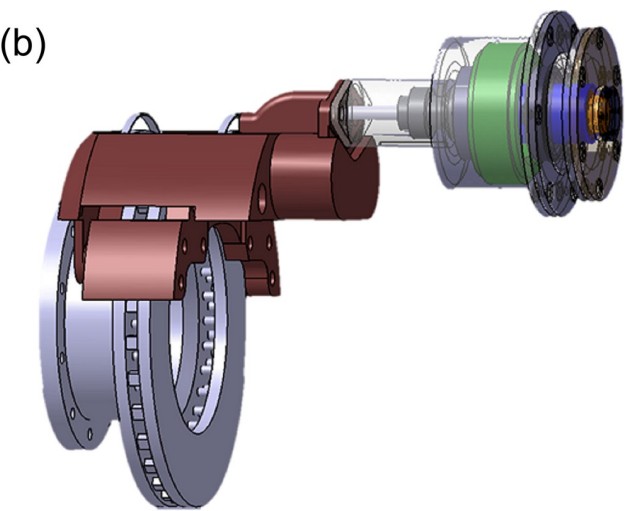

**Fig 2. EMB of electric bus.** (a) Structure and working principle: (1) motor stator, (2) motor rotor, (3) head, (4) screw, (5) force sensor, (6) push rod, (7) linear sensor, (8) arm, (9) piston, (10) brake shoes, (11) brake disc, (12) caliper. (b) 3D model.

the maximum clamping force in most EMB schemes cannot meet the needs of commercial vehicles. Therefore, it is important to design a simple, flexible, and versatile EMB scheme and build a corresponding clamping force control strategy for commercial vehicles.

In this study, an EMB scheme for electric buses based on pneumatic disc brake is established, and a corresponding clamping force control strategy is proposed. The proposed EMB consists of a powerplant and brake caliper, as shown in Fig 2. Particularly, to retain the pneumatic brake caliper, the brake chamber is replaced by the powerplant to push the mechanism inside the caliper that generates the clamping force. Compared with the existing EMB schemes, the proposed EMB is driven by the motor directly, and caliper is equipped with an automatic clearance adjustment device; hence, it has the advantages of direct transmission and simple control as that of scheme A (shown in Fig 1a). In addition, the arm inside the caliper can enlarge the axial thrust force output from the powerplant; thus, the output capacity of the EMB is improved, and the performance requirements of the motor are reduced. The proposed

EMB satisfies the brake demand of commercial vehicles and overcomes the control difficulty and high nonlinearity of scheme B (shown in Fig 1b). Furthermore, the pneumatic brake caliper is retained in the proposed EMB, making it simple, flexible, and versatile; this scheme can be conveniently transferred from pneumatic brake to EMB. However, we observed a problem during the process of research: although the force arm inside the caliper can magnify the axial thrust force output from the powerplant and act on the brake shoes, it enlarges the axial displacement of the brake shoes into the screw, according to the leverage principle (shown in Fig 2a). Limited by the ultimate performance of the motor, it is time–consuming to eliminate brake clearance, which affects the EMB performance and driving safety, which is unfavorable.

To minimize the scheme's adverse effects, the primary focus of this study is to improve the clamping force response performance without changing the structure and size of the EMB. In recent years, with increasing interdisciplinary integration and development of computer technology, it is possible to combine advanced technology, such as pattern recognition and machine learning, with motion control of the mechanical system. Therefore, this study aims to combine the driver intentions with the EMB clamping force control, accurately identify the driver intentions based on existing sensor signals, and eliminate the brake clearance appropriately in advance. Finally, the study aims to shorten the overall clamping force response time and minimize the adverse effect caused by the leverage effect in the proposed EMB.

This paper is organized as follows: an EMB scheme for an electric bus is proposed, and the corresponding system model is deduced in Section 2; the HMMs of driver decelerate, and brake intentions from the relationship between driving conditions, driver intentions, and pedal behaviors are established in Section 3; parameters of the HMMs are determined, and driver intentions are recognized in Section 4; clamping force control strategy and simulation analyses are performed in Section 5; conclusions are made in Section 6.

## 2. System modeling

This section briefly introduces the proposed EMB structure, succinctly describes its working principle, and establishes its system model.

### 2.1 EMB structure and working principle

The EMB is generally composed of a powerplant and brake caliper. The powerplant includes a brushless DC (BLDC) motor, screw mechanism, and push rod integrated with the screw.

The power transmission path of the EMB is shown with an arrow in Fig 2a. During the working of the system, the motor stator (1) is fixed, motor rotor (2) is connected with the head (3), and screw (4) transforms the torque output from the BLDC motor into axial thrust force on the push rod (6) and act on the arm (8) in the caliper. The lower end of the arm exerts an amplified force on the piston (9) and drives the piston to push the brake shoes (10) to press the brake disc (11) for generating the clamping force. The pressure sensor (5) measures the clamping force, while the linear sensor (7) measures the distance axially traveled by the screw.

### 2.2 EMB system modeling

The EMB is a complex mechanical system, and the accuracy of its model affects the control performance. Therefore, according to the scheme, the EMB system model is divided into two parts: powerplant, and caliper. The powerplant model is composed of a BLDC motor and screw mechanism, fabricated according to the electrical fundamentals and kinematics. The caliper model is complex and described using manufacturer specifications and fitted in this study to ensure accuracy and reduce complexity.

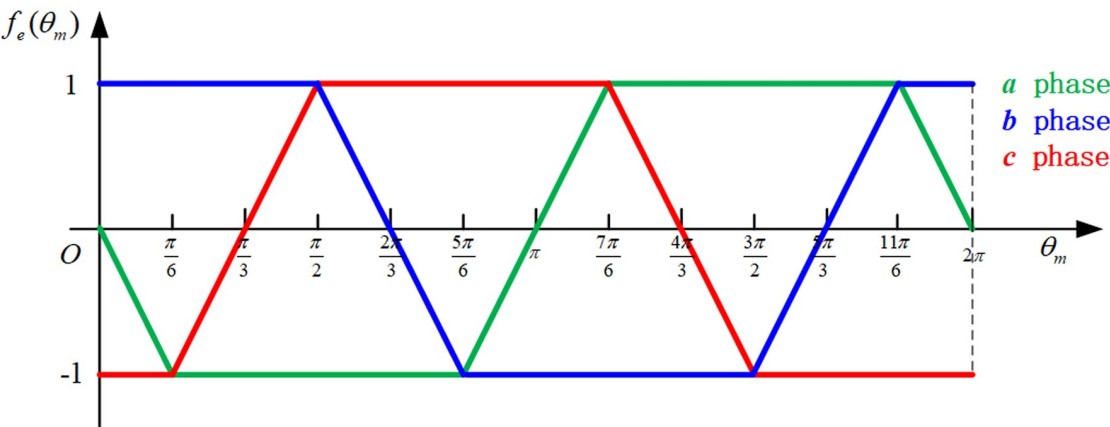

**Fig 3. Back EMF waveform of BLDC.**

**2.2.1 Crucial part: Powerplant.** The powerplant axially extends the screw under the BLDC motor drive, and convert the motor torque into an axial thrust force.

1. BLDC motor
According to Ohm's law, the equation of the three-phase terminal voltage is given by Eq (1) [25].

$$
\begin{bmatrix} u_a \\ u_b \\ u_c \end{bmatrix} = \begin{bmatrix} R & 0 & 0 \\ 0 & R & 0 \\ 0 & 0 & R \end{bmatrix} \begin{bmatrix} i_a \\ i_b \\ i_c \end{bmatrix} + \begin{bmatrix} L-M & 0 & 0 \\ 0 & L-M & 0 \\ 0 & 0 & L-M \end{bmatrix} \frac{d}{dt} \begin{bmatrix} i_a \\ i_b \\ i_c \end{bmatrix} + \begin{bmatrix} e_a \\ e_b \\ e_c \end{bmatrix} \tag{1}
$$

where $u$ is the three-phase terminal voltage of the stator; $R$ is the phase resistance; $i$ and $e$ are the three-phase current and back EMF, respectively; $L$ and $M$ are the self and mutual inductance of the three-phase windings, respectively. Electromagnetic torque output $T_e$ of the motor is as follows:

$$
T_e = p\psi_m[f_a(\theta_m)i_a + f_b(\theta_m)i_b + f_c(\theta_m)i_c] \tag{2}
$$

where $p$ is the pole pairs of the motor; $\psi_m$ is maximum winding flux; $f(\theta_m)$ is the back EMF waveform, as shown in Fig 3.

2. Screw mechanism
The screw mechanism is composed of an active component head (3) and a follower screw (4). Fig 4 is the schematic diagram of the force analyses after the circumferential expansion of the screw mechanism [7]. During the working of the mechanism, the head is rotated under the input torque $T_e$ and drives the screw axially.
According to Fig 4,

$$
\tan(\alpha + \rho) = \frac{l}{\pi d_m} \tag{3}
$$

where $\alpha$ and $\rho$ are the helix angle and equivalent friction angle of the screw, respectively, and $l$ and $d_m$ are the lead range and nominal diameter of the screw, respectively.
If the friction between the screw and head is ignored, the relationship between the input

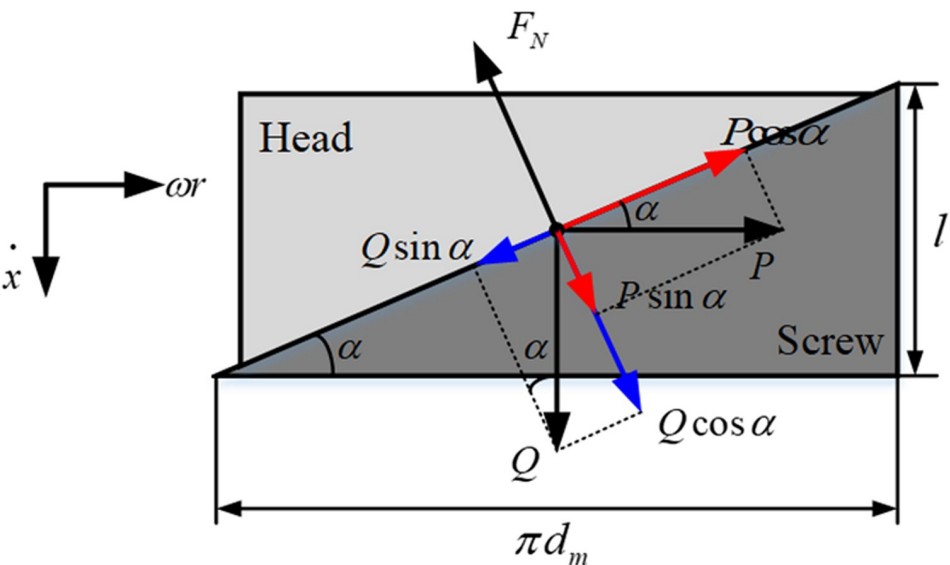

**Fig 4. Force analyses of the threaded mechanism.**

torque $T_e$ and axial thrust force $Q$, output from the screw, is as follows [7]:

$$T_e\eta = Q\frac{d_m}{2}\tan(\alpha + \rho) \tag{4}$$

where $\eta$ is the efficiency of the screw mechanism. Force $Q$ can be described as:

$$Q = \frac{2\pi\eta \cdot T_e}{l} \tag{5}$$

Thus, the clamping force is:

$$F_{cl} = kQ \tag{6}$$

where $k$ is the force amplifying coefficient of the caliper. The axial travel of the screw is as follows:

$$x_a = \frac{\theta_m \cdot l}{2\pi} \tag{7}$$

where $\theta_m$ is the rotation angle of the head (and the rotation angle of the rotor).

3. Powerplant model

According to Newton's second law, the motion equation of the powerplant can be obtained as follows:

$$T_t - T_f - T_L = J\frac{d\omega_m}{dt} \tag{8}$$

where $T_t$ and $T_L$ are the output and load torques of the powerplant, respectively; $T_f$ is the friction torque of the motor; $J$ is the equivalent moment inertia to the rotor; $\omega_m$ is the rotor

mechanical angular velocity. Further:

$$T_t = \eta \cdot T_e \tag{9}$$

$$T_f = D\omega_m \tag{10}$$

$$T_L = \frac{F_L \cdot d_m}{2k} \tag{11}$$

where $D$ is the viscous damping coefficient of the motor; $F_L$ is the reaction force between brake shoes and disc.

If $i_a$, $i_b$, $i_c$, and $\omega_m$ are selected as the system state variables, $u_a$, $u_b$, $u_c$, and $F_L$ as the system inputs, and clamping force $F_{cl}$ and screw axial velocity $v_x$ as the outputs, then the power-plant model can be expressed as follows:

$$\begin{cases} \dot{x} = Ax + Bu \\ y = Cx \end{cases} \tag{12}$$

where,

$$x = (i_a, i_b, i_c, \omega_m)^T, y = (F_{cl}, v_x)^T, u = (u_a, u_b, u_c, F_L)^T,$$

$$A = \left( \begin{array}{ccc|c} -R/(L-M) & 0 & 0 & -p\psi_m f_a(\theta_m)/(L-M) \\ 0 & -R/(L-M) & 0 & -p\psi_m f_b(\theta_m)/(L-M) \\ 0 & 0 & -R/(L-M) & -p\psi_m f_c(\theta_m)/(L-M) \\ \hline \eta p\psi_m f_a(\theta_m)/J & \eta p\psi_m f_b(\theta_m)/J & \eta p\psi_m f_c(\theta_m)/J & -D/J \end{array} \right),$$

$$B = \left( \begin{array}{ccc|c} 1/(L-M) & 0 & 0 & 0 \\ 0 & 1/(L-M) & 0 & 0 \\ 0 & 0 & 1/(L-M) & 0 \\ \hline 0 & 0 & 0 & -d_m/2kJ \end{array} \right),$$

$$C = \left( \begin{array}{ccc|c} 2\pi\eta p\psi_m f_a(\theta_m)/l & 2\pi\eta p\psi_m f_b(\theta_m)/l & 2\pi\eta p\psi_m f_c(\theta_m)/l & 0 \\ 0 & 0 & 0 & l/2\pi \end{array} \right).$$

### 2.2.2 Crucial part: Caliper.
There are two main functions of the caliper:

1. To amplify the axial thrust force output from the powerplant and transform it into the clamping force through the action between the brake shoes and disc;

2. Simultaneously, the caliper applies the reaction force to the powerplant, according to Newton's third law, for the EMB system to attain a balanced state. At this point, the caliper is equivalent to a "load device".

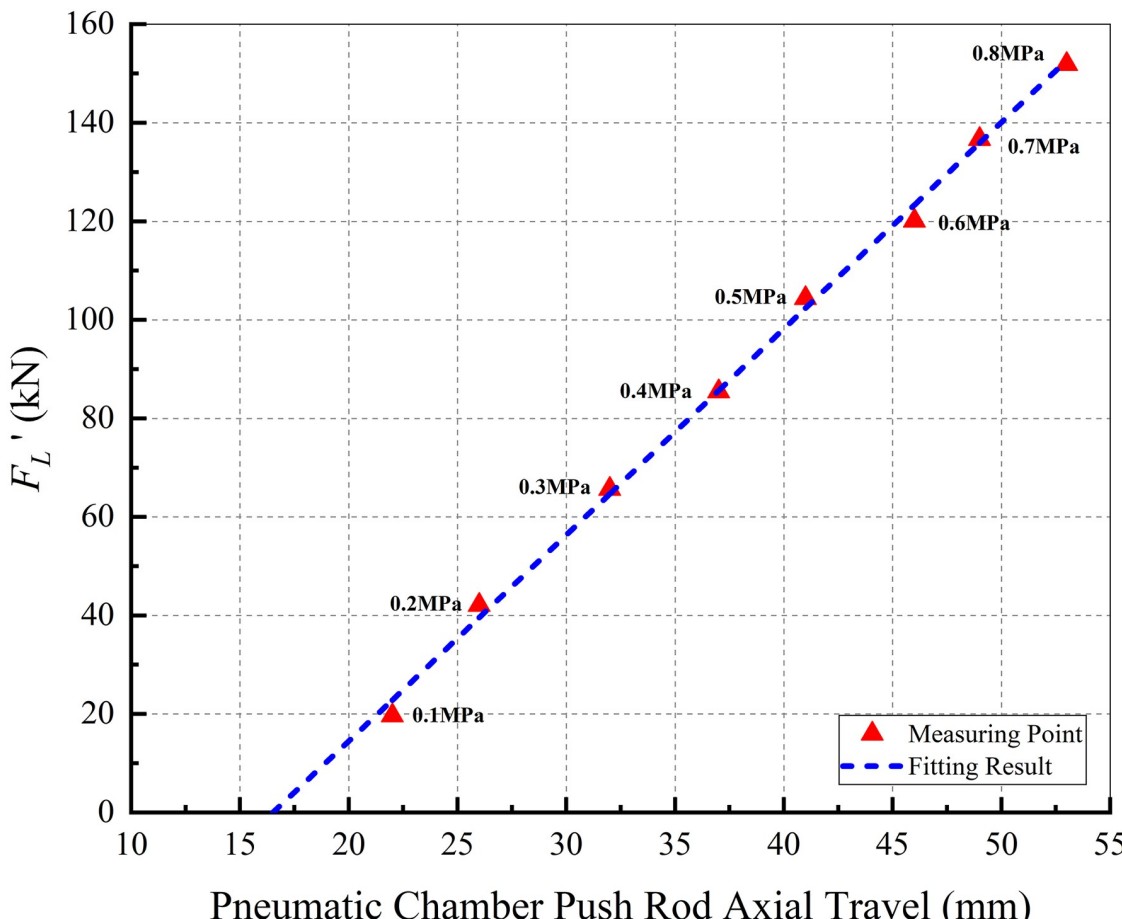

**Fig 5. Characteristic of the caliper.**

Therefore, the caliper model should accurately reflects the "load". In this study, the characteristic of a pneumatic disc brake is used for the caliper model, which describes the relationship between $x'$ and $F'_L$. Here, $x'$ is the axial travel of chamber push rod and $F'_L$ is the force between brake shoes and disc. According to the pneumatic brake working principle, $x'$ corresponds to the screw axial travel distance $x_a$ in the proposed EMB, and $F'_L$ is equal to the load force $F_L$ of the caliper acting on the powerplant. Fig 5 shows the characteristic of a pneumatic disc brake; a series of discrete points in the figure represent the measured data provided by the caliper manufacturer, and the blue dotted line is the fitted characteristic curve. The caliper model is further observed in the figure.

**2.2.3 Integrated: EMB system model.**　The EMB system model is shown in Fig 6. It can be observed that the inputs of the model are $u_a$, $u_b$, $u_c$, and $F_L$, where $u_a$, $u_b$, $u_c$ are obtained from the outside and $F_L$ is determined by the curve, as shown in Fig 5. The model outputs are the clamping force $F_{cl}$ and screw axial velocity $v_x$, where the screw axial travel $x_a$ can be obtained through integration. The key parameters of the EMB system model are listed in Table 1.

## 3. Modeling of driver decelerate and brake intentions

In this section, the relationship between driving conditions, driver intentions, pedal behaviors, and HMMs of driver's decelerate and brake intentions is established. It is assumed that the

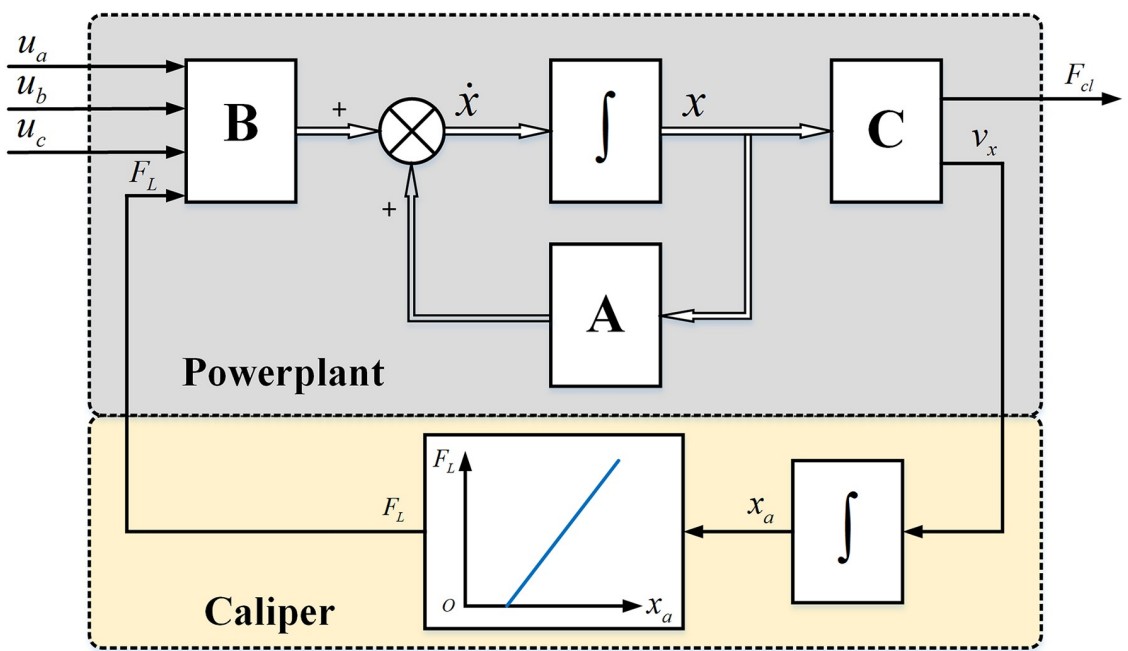

**Fig 6. EMB system model.**

driver controls the vehicle using the accelerator or brake pedal and not with the steering wheel.

## 3.1 Relationship among driving conditions, driver intentions, and pedal behaviors

If the steering behavior is ignored, the driving conditions mainly include acceleration, deceleration, sliding, and braking. In the driving process, the driver pedal behaviors reflect the intentions and correspond to the driving conditions. The intentions represent the driver's strategy to operate the vehicle. Particularly, when the driver wishes to decelerate, he releases the accelerator pedal first and then appropriately decides to hit the brake pedal. If the driver's decelerate and brake intentions are divided into three levels, namely mild, moderate, and strong intentions, then the velocity of releasing the accelerator pedal or hitting the brake pedal reflects the intensities of driver's decelerate or brake intentions, respectively.

For an electric vehicle, the relationship between the driving conditions, driver intentions, and pedal behaviors is listed in Table 2.

The driver intentions to accelerate, decelerate, slide, or brake can be preliminarily determined according to the table. For the deceleration and braking conditions, the driver

**Table 1. Key parameters of the EMB system model.**

| Symbol | Meaning | Value | Symbol | Meaning | Value |
|--------|---------|-------|--------|---------|-------|
| $R$ | phase resistance | $0.43\ \Omega$ | $J$ | equivalent moment inertia to rotor | $3.0879 \times 10^{-3}\ \mathrm{kg \cdot m^2}$ |
| $L$ | self-inductance | $10.2\ \mathrm{mH}$ | $D$ | viscous damping coefficient of motor | $1 \times 10^{-3}\ \mathrm{N \cdot m \cdot s}$ |
| $M$ | mutual inductance | $1.6\ \mathrm{mH}$ | $p$ | pole pairs of motor | $10$ |
| $l$ | screw lead range | $6\ \mathrm{mm}$ | $\psi_m$ | maximum winding flux | $0.0225\ \mathrm{Wb}$ |
| $d_m$ | screw nominal diameter | $32\ \mathrm{mm}$ | $k$ | force amplifying coefficient of the caliper | $15.2$ |

**Table 2. Relationship among vehicle driving conditions, driver's intentions and pedal behaviors.**

| Conditions | Intentions | Pedal behaviors |
|---|---|---|
| acceleration | accelerate intention | hit the accelerator pedal |
| deceleration | mild, moderate, or strong decelerate intentions | release the accelerator pedal |
| sliding | slide intention | without pedal operation |
| braking | mild, moderate, or strong brake intentions | hit or release the brake pedal |

intentions are further judged based on the models established below. The acceleration and sliding conditions are beyond the scope of this study. In addition, unless specified, the driver decelerate and brake intentions are collectively referred to as the driver intentions in further sections.

## 3.2 Establishing the HMMs

The driver intentions are affected by factors such as road conditions, vehicle status, weather conditions, and self–cause during actual driving, which exhibit randomness and are time–varying. When analyzed, the forming of driver intention can be regarded as a stochastic process. When compared with the measurable signals, such as pedal travel, pedal velocity, and pedal force, the driver intentions are difficult to be quantified. Hence, the purpose of the driver intentions recognition is to describe such ambiguous intentions using measurable signals.

As the driver intentions at each moment are relatively independent, and the intention $S_t$ at moment $t$ is solely related to the intention $S_{t-1}$ of the previous moment $t$-1, the Markov property is exhibited. Therefore, the driver intentions in a short time can be regarded as a Markov chain, which is given by:

$$P(S_t|S_{t-1}, S_{t-2}, \ldots, S_1) = P(S_t|S_{t-1}) \tag{13}$$

To demonstrate the mathematical relationship between the driver intentions and pedal behaviors, the HMMs of the driver's decelerate and brake intentions are established in the following sections.

**3.2.1 Model structure and hypothesis.** The HMM is a probabilistic time–series model used to describe a process where a Markov chain with unobservable random states sequence generates an observable random sequence, as shown in Fig 7. A complete HMM consists of hidden states $S$ and observed states $O$ and is described by the number of hidden states $m$, number of observed states $n$, initial probability matrix of hidden states $\pi$, one–step transition probability matrix $A$ of hidden states, and emission probability matrix $P$ from the hidden states to observed states, i.e.,

$$\lambda = \{m, n, \pi, A, P\} \tag{14}$$

Prior to establishing HMMs of the driver intentions, the following assumptions were made:

1. Immobility hypothesis, i.e., assuming driver intentions to be independent of the specific moment, can be described as: $P(S_{m+1} = j|S_m = i) = P(S_{n+1} = j|S_n = i)$, $\forall m,n$;

2. Output independence hypothesis, i.e., the output (observed state) of the HMM is solely related to the current driver intention, is given by: $P(O_1, O_2, \ldots, O_T|S_1, S_2, \ldots, S_T) = \prod P(O_t|S_t)$, $t = 1, 2, \ldots, T$;

3. Transfer adjacency hypothesis, i.e., the direct transition of the hidden state between two non–adjacent hidden states is not possible.

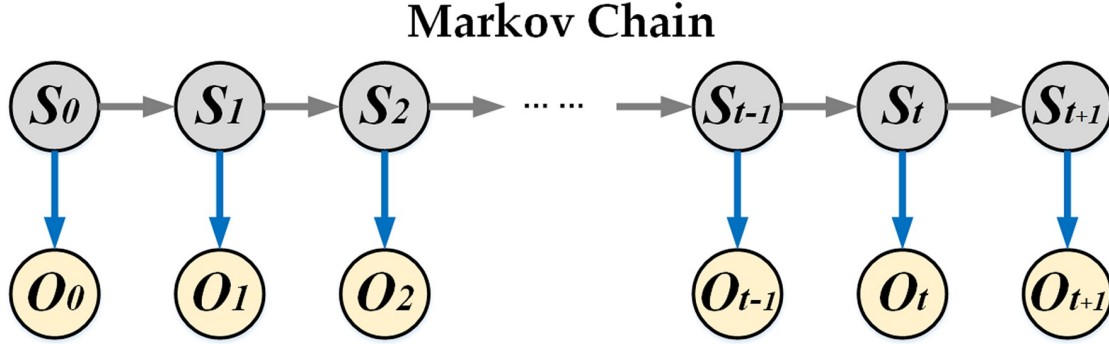

**Fig 7. Hidden Markov model structure.**

**3.2.2 Hidden states $S$, initial probability matrix $\pi$ and one-step transition probability matrix $A$.** It is difficult to quantify the driver intentions, that possess hidden characteristics, when driving; these intentions are determined as hidden states of the model. More the number of hidden states, the higher the model accuracy. However, the HMM involves numerous matrix operations, and several hidden states affect the efficiency of the model. Considering the actual driving situation, the numbers of hidden states in the proposed HMM models are set to three, indicating the driver's mild, moderate, and strong intention.

The driver intention at the initial moment is described by the initial probability matrix $\pi$:

$$\pi = (\pi_1, \pi_2, \cdots, \pi_m)_{1m}^{1 \times m1m} \qquad (15)$$

where $\pi_i$ ($i = 1, 2, \ldots, m$) represents the probability that the driver intention is in state $i$ at the initial moment $t = 1$.

The probability of the driver intentions transferring from one state to another is described by the one–step transition probability matrix $A = (a_{ij})_{m \times m}$, and $a_{ij} = P(S_{t+1} = j \mid S_t = i)$ represents the probability that the driver intention at moment $t$ in state $i$ transferring to the state $j$ at moment $t+1$ is $a_{ij}$.

**3.2.3 Observed states $O$.** The most intuitive indicator of pedal behaviors is force. However, most vehicles are not equipped with a pedal force sensor. Hence, the pedal velocity is selected as the observed state of the HMMs to describe the driver pedal behaviors. According to the aforementioned analyses, this study determines the number of observed states $n_1 = 4$ part of the driver decelerate intention HMM $\lambda_1$, i.e., the driver holds the accelerator pedal, and releases the accelerator pedal slowly, moderately, and quickly. Similarly, the number of observed states $n_2 = 7$ part of the driver brake intention HMM $\lambda_2$ represents the situation when the driver hits brake pedal slowly, moderately, and quickly, holds the brake pedal, and releases the brake pedal slowly, moderately, and quickly.

**3.2.4 Emission probability matrix $P$.** The observed states of the HMMs (i.e., the different pedal velocities) depend on the hidden states (i.e., the driver intentions), and the relationship between hidden states and observed states is described by the emission probability matrix $P = (p_{ik})_{m \times n}$, $i = 1, 2, \ldots, m$, $k = 1, 2, \ldots, n$.

Here, $p_{ik} = P(O_t = k \mid S_t = i)$ indicates the probability that the driver intention $S_t$ in state $i$ at moment $t$ and pedal velocity $O_t$ in state $j$ at the same moment is $p_{ik}$.

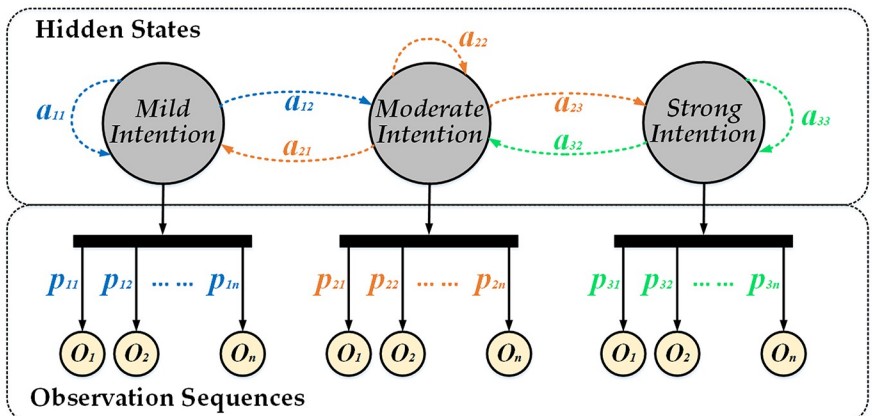

**Fig 8. HMM of driver intentions.**

**3.2.5 HMMs of driver's decelerate and brake intentions.** The HMMs of the driver intentions are shown in Fig 8. In addition, the elements in each HMM are listed in Table 3.

# 4. Determining HMM parameters and recognizing driver intentions

In this section, the pedal velocity data are discretized and segmented into several sections to establish the relationship between pedal velocities and observed states. Then, the HMM parameters are determined using AHP and MATLAB HMM toolbox. Finally, the driver's decelerate and brake intentions are recognized using the Viterbi algorithm.

## 4.1 Determining HMM parameters

The HMM parameters are determined as follows: For the given observed state sequence $\{O_1, O_2,\ldots,O_t\}$, initial probability matrix $\pi$, estimated transition probability matrix $\hat{A}$, and emission probability matrix $\hat{P}$ as the inputs, the HMM parameters with the maximum probability for the given observed state sequence can be obtained by iterating solutions using the Baum–Welch algorithm.

**4.1.1 Observed state sequence.** To obtain the observed state sequence, the pedal velocity signals are discretized into several sections are then coded (the number of sections is consistent with the number of observed states $n$). The observed state of the HMM is then matched to the pedal velocity to obtain the sequence.

The accelerator and brake pedal velocities were obtained by acquiring actual driving data. The accelerator pedal velocity is discretized into four sections (corresponding to the four observed states in the decelerate intention HMM), and brake pedal velocity is discretized into seven sections (corresponding to the seven observed states in the brake intention HMM). For

**Table 3. Elements in the HMMs of driver intentions.**

| HMMs | Hidden states | Observed states | Initial probability | Transition probability | Emission probability |
|---|---|---|---|---|---|
| Decelerate intention $\lambda_1$ | $m_1 = 3$ | $n_1 = 4$ | $(\pi_1)_{1\times3}$ | $(A_1)_{3\times3}$ | $(P_1)_{3\times4}$ |
| Brake intention $\lambda_2$ | $m_2 = 3$ | $n_2 = 7$ | $(\pi_2)_{1\times3}$ | $(A_2)_{3\times3}$ | $(P_2)_{3\times7}$ |

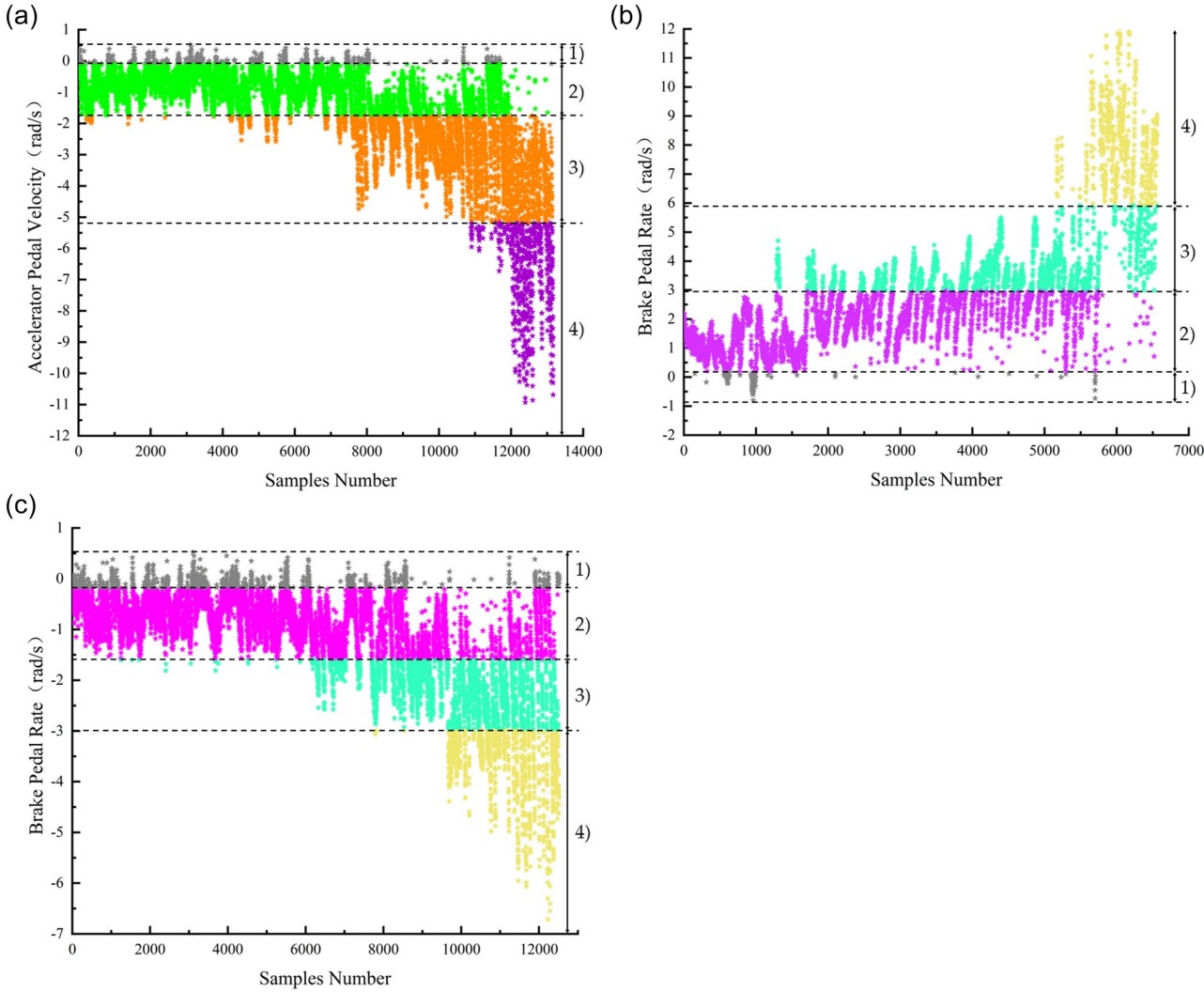

**Fig 9. Pedal velocity discretizing results.** (a) Release the accelerator pedal. (b) Hit the brake pedal. (c) Release the brake pedal.

the pedal velocity at moment $t$ located at the $k^{th}$ ($k = 1, 2,. . ., n$) section, the observed state of the HMMs is $O_t = k$.

The discretion of the pedal velocity is based on the k-means algorithm. The basic principle of k-means is to move all kinds of data centers and redistribute various data members continuously, and shift the position of different centers to the average value of their members so that the members of the same class are closest to their corresponding centers. The pedal velocity discretizing results are shown in Fig 9. When the accelerator pedal reaches its maximum travel, its angle is approximately 0.29 rad (approximately 17˚), and the angle of the brake pedal is approximately 0.69 rad (approximately 40˚).

Upon analysis of the results shown in Fig 9, the corresponding relationship between the observed states and pedal velocities is obtained, as shown in Table 4.

**4.1.2 Initial probability matrix $\pi$ of hidden states.** Assuming that the probability of driver intention in every hidden state is equal at the initial moment, the initial probability

Table 4. Corresponding relationship between observed states and pedal velocities.

| HMMs | Observed States | Meaning of observed states (pedal behaviors) | Pedal Velocities (rad/s) | Discretized sections |
|---|---|---|---|---|
| Decelerate Intention HMM | 1 | hold the accelerator pedal | [−0.1, 0.1] | Fig 9(a)-1) |
| | 2 | release the accelerator pedal slowly | [−1.74, −0.1] | Fig 9(a) -2) |
| | 3 | release the accelerator pedal moderately | [−5.18, −1.74) | Fig 9(a) -3) |
| | 4 | release the accelerator pedal quickly | (−∞, −5.18) | Fig 9(a) -4) |
| Brake Intention HMM | 1 | hit the brake pedal slowly | (0.2, 2.97] | Fig 9(b) -2) |
| | 2 | hit the brake pedal moderately | (2.97, 5.93] | Fig 9(b) -3) |
| | 3 | hit the brake pedal quickly | (5.93, +∞) | Fig 9(b) -4) |
| | 4 | hold the brake pedal | [−0.2, 0.2] | Combine Fig 9(b) - 1) and Fig 9(c) - 1) |
| | 5 | release the brake pedal slowly | [−1.57, −0.2) | Fig 9(c) -2) |
| | 6 | release the brake pedal moderately | [−2.97, −1.57) | Fig 9(c) -3) |
| | 7 | release the brake pedal quickly | (−∞, −2.97) | Fig 9(c) -4) |

matrix $\pi$ can be expressed as

$$\pi = \begin{pmatrix} 1/3 & 1/3 & 1/3 \end{pmatrix} \tag{16}$$

**4.1.3 Estimated transition probability matrix $\hat{A}$.** $f_{ij}$ $(i, j = 1, 2, \ldots, m)$ is defined as the times that the driver intention transferred from state $i$ to state $j$ directly. According to the 3$^{rd}$ assumption and Fig 8, $f_{ij}$ is as follows:

$$f_{ij} = \begin{pmatrix} 1 & 1 & 0 \\ 1 & 1 & 1 \\ 0 & 1 & 1 \end{pmatrix} \tag{17}$$

In Markov chain, the maximum likelihood estimator of transition probability is:

$$\hat{a}_{ij} = \frac{f_{ij}}{\sum\limits_{k=1}^{m} f_{ik}} \quad (i, j = 1, 2, \ldots, m) \tag{18}$$

Therefore, the estimated transition probability matrix $\hat{A}$ can be described as:

$$\hat{A} = \begin{pmatrix} 1/2 & 1/2 & 0 \\ 1/3 & 1/3 & 1/3 \\ 0 & 1/2 & 1/2 \end{pmatrix} \tag{19}$$

Since $f_{ij}$ of driver intentions are equal, the matrix $\hat{A}$ for the two HMMs will be equal.

**4.1.4 Estimated emission probability matrix $\hat{P}$.** The estimated emission probability matrix $\hat{P}$ affects the accuracy of the HMM model to a certain extent [26], and the reliability of $\hat{P}$ determined through experience remains to be tested. In this study, $\hat{P}$ is determined using the analytic hierarchy process (AHP) method. The AHP method is mainly composed of the following steps:

Step 1. Establish the hierarchical analysis model
The focus of AHP is to establish a hierarchical analysis model based on the relationship between the driver intentions and pedal behaviors, as shown in Fig 10.

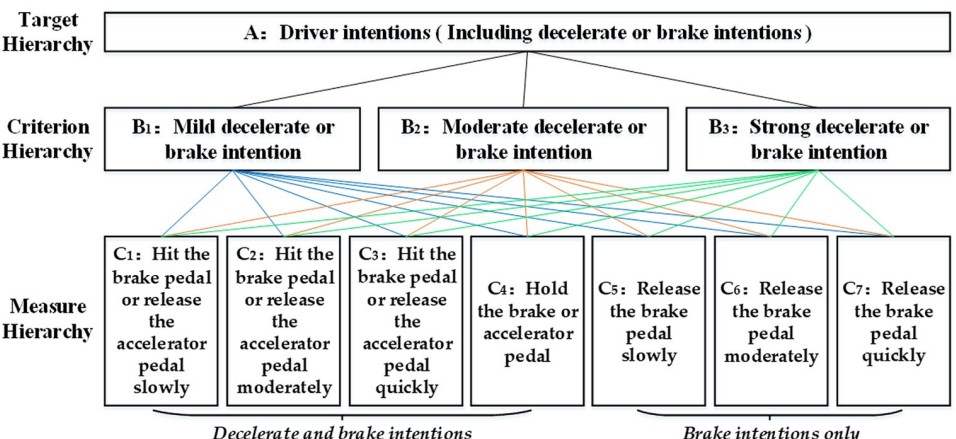

**Fig 10. Hierarchical analysis model of driver intentions.**

In this model, the target hierarchy $A$ is the driver intentions, particularly, the driver's decelerate or brake intention; the target hierarchy $A$ is further separated into criterion hierarchy $B$, that include the mild decelerate or brake intention $B_1$, moderate decelerate or brake intention $B_2$, and strong decelerate or brake intention $B_3$; the measure hierarchy $C$, include different pedal behaviors $C_1$—$C_7$, to achieve the intentions involved in hierarchy $B$. Among them, $B_1$—$B_3$ are called the elements and $C_1$—$C_7$ are called the factors.

Step 2. Construct the judgment matrix

The judgment matrix $K_p$ is used to compare the mutual influence degree of various factors on a certain element. The principle to determine the judgment matrix $K_p$ is to compare all the factors with each other, and establish the judgment scale $k_{ij}$ according to their mutual influence on a certain element to construct the judgment matrix $K_p = (k_{ij})_{n \times n}$.

Here, the range of judgment scale $k_{ij}$ is 1–9, scale '1' means that the factors $C_i$ ($i = 1, 2, \ldots, 7$) and $C_j$ ($j = 1, 2, \ldots, 7$) are equally important for the element $B_p$, and scale '9' means that the factor $C_i$ is extremely important than $C_j$. If the influence degree of factor $C_i$ and $C_j$ on element $B_p$ is $k_{ij}$, then the influence degree of factor $C_j$ and $C_i$ on element $B_p$ is $k_{ji} = 1/k_{ij}$.

If the influence of accelerator pedal behaviors $C_1$—$C_4$ on driver mild decelerate intention $B_1$ is considered as an example, the judgment matrix $K_{1,dec}$ is constructed as follows:

$$K_{1,dec} = \begin{pmatrix} 1 & 3 & 5 & 7 \\ 1/3 & 1 & 3 & 5 \\ 1/5 & 1/3 & 1 & 3 \\ 1/7 & 1/5 & 1/3 & 1 \end{pmatrix} \tag{20}$$

It can be seen from the above analyses that three judgment matrices each can be constructed under the driver's decelerate and brake intentions, as listed in S1 Appendix.

Step 3. Calculate influence weight

Unlike the judgment matrix $K_p$, the influence weight $H_i$ is used to describe the influence degree of factors on each element. The influence weight $H_i$ is calculated using the eigenvalue method and normalized, i.e., the eigenvalues $\lambda = (\lambda_1, \lambda_2, \ldots, \lambda_n)$ of each judgment matrix $K_p$ are calculated, and the eigenvector $\omega = (\omega_1, \omega_2, \ldots, \omega_n)^T$ corresponding to the maximum eigenvalue $\lambda_{max}$ can be used as the influence coefficient $h_i$ of a factor $C_i$ ($i = 1$,

2,..., n) on an element $B_p$ after normalization, i.e.,

$$h_i = \frac{w_i}{\sum\limits_{k=1}^{n} w_k}$$ (21)

The row vector $H = (h_1, h_2, \ldots, h_n)$ composed of all influence coefficient $h_i$ which is the influence weight of factors $C_1, C_2, \ldots, C_n$ on an element $B_p$. For the hierarchical analysis model of the driver decelerate intentions, the influence weights $H_{dec}$ of factors (accelerator pedal behaviors) on each element (driver deceleration intention) are as follows:

$$H_{1,dec} = (0.5650, 0.2622, 0.1175, 0.0553);$$
$$H_{2,dec} = (0.0603, 0.1155, 0.6223, 0.2019);$$ (22)
$$H_{3,dec} = (0.0563, 0.1310, 0.2388, 0.5738).$$

Similarly, for the hierarchical analysis model of the driver brake intentions, the influence weights $H_{brk}$ of factors (brake pedal behaviors) on each element (driver brake intention) are as follows:

$$H_{1,brk} = (0.2458, 0.0653, 0.0242, 0.0611, 0.0968, 0.1577, 0.3490);$$
$$H_{2,brk} = (0.0671, 0.2479, 0.0683, 0.1301, 0.1685, 0.2805, 0.0375);$$ (23)
$$H_{3,brk} = (0.0471, 0.1553, 0.3092, 0.1747, 0.1896, 0.0764, 0.0477).$$

Step 4. Determine the emission probability matrix

As the influence coefficient $h_i$ is normalized, the definition of influence coefficient $h_i$ and influence weight $H_i$ can be combined, and the matrix composed of influence weights $H_1$, $H_2$, $H_3$ under the same intention can be used as the estimated emission probability matrix $\hat{P}$ of the HMMs. Therefore, the estimated emission probability matrix $\hat{P}_1$ of the driver decelerate intention HMM is:

$$\hat{P}_1 = \begin{pmatrix} H_{1,dec} \\ H_{2,dec} \\ H_{3,dec} \end{pmatrix} = \begin{pmatrix} 0.5650 & 0.2622 & 0.1175 & 0.0553 \\ 0.0603 & 0.1155 & 0.6223 & 0.2019 \\ 0.0563 & 0.1310 & 0.2388 & 0.5738 \end{pmatrix}$$ (24)

Similarly, the estimated emission probability matrix $\hat{P}_2$ of the driver brake intention HMM is:

$$\hat{P}_2 = \begin{pmatrix} H_{1,brk} \\ H_{2,brk} \\ H_{3,brk} \end{pmatrix} = \begin{pmatrix} 0.2458 & 0.0653 & 0.0242 & 0.0611 & 0.0968 & 0.1577 & 0.3490 \\ 0.0671 & 0.2479 & 0.0683 & 0.1301 & 0.1685 & 0.2805 & 0.0375 \\ 0.0471 & 0.1553 & 0.3092 & 0.1747 & 0.1896 & 0.0764 & 0.0477 \end{pmatrix}$$ (25)

Step 5. HMM parameters training

The model parameters training uses Baum-Welch algorithm and MATLAB HMM toolbox. According to Table 4, the collected pedal velocity data with different driver intentions are transformed into the observed state sequence of HMMs and imported into the MATLAB HMM toolbox. Then, the observed state sequence, initial probability matrix $\pi$ of hidden states, estimated transition probability matrix $\hat{A}$, and emission probability matrix $\hat{P}$ are

**Table 5. Trained parameters of the HMMs.**

| Driver decelerate intention HMM $\lambda_1 = (m_1, n_1, \pi_1, A_1, P_1)$ | Driver brake intention HMM $\lambda_2 = (m_2, n_2, \pi_2, A_2, P_2)$ |
|---|---|
| $m_1 = 3$ | $m_2 = 3$ |
| $n_1 = 4$ | $n_2 = 7$ |
| $\pi_1 = ( 1/3 \quad 1/3 \quad 1/3 )$ | $\pi_2 = ( 1/3 \quad 1/3 \quad 1/3 )$ |
| $A_1 = \begin{pmatrix} 0.9953 & 0.0047 & 0 \\ 0.0034 & 0.9905 & 0.0061 \\ 0 & 0.0110 & 0.9890 \end{pmatrix}_{3\times3}$ | $A_2 = \begin{pmatrix} 0.9843 & 0.0157 & 0 \\ 0.0125 & 0.9768 & 0.0107 \\ 0 & 0.0069 & 0.9931 \end{pmatrix}_{3\times3}$ |
| $P_1 = \begin{pmatrix} 0.8127 & 0.1873 & 0 & 0 \\ 0.0367 & 0.9459 & 0.0174 & 0 \\ 0.0005 & 0.0449 & 0.8173 & 0.1373 \end{pmatrix}_{3\times4}$ | $P_2 = \begin{pmatrix} 0.4115 & 0.0262 & 0.0001 & 0.0188 & 0.0200 & 0.3581 & 0.1653 \\ 0.0085 & 0.2328 & 0.0010 & 0.0912 & 0.6358 & 0.0307 & 0 \\ 0.0077 & 0.0003 & 0.0545 & 0.9064 & 0.0311 & 0 & 0 \end{pmatrix}_{3\times7}$ |

used as inputs for the HMM parameters training. The training is completed when the calculation error is less than $10^{-6}$. The trained parameters of the HMMs are shown in Table 5.

## 4.2 Driver decelerate and brake intention recognition

The driver intention recognition is as follows: Using the Viterbi algorithm, for the given HMM parameters, an optimal hidden state sequence $\{S_1, S_2, \ldots, S_t\}$ is calculated, which can best explain the observed state sequence $\{O_1, O_2, \ldots, O_t\}$.

**4.2.1 Viterbi algorithm.** If $S_1^t$ refers to the hidden state sequence $\{S_1, S_2, \ldots, S_t\}$, $O_1^t$ refers to the observed state sequence $\{O_1, O_2, \ldots, O_t\}$, $s_1^t$ represents the value of the hidden state sequence $S_1^t$, and $o_1^t$ represents the value of the observed state sequence $O_1^t$, then, the Viterbi algorithm maximizes the probability $P(S_1^t = s_1^t, O_1^t = o_1^t)$. In this algorithm, the Viterbi matrix is created, and the Viterbi formula is used to find the optimal hidden state sequence. Viterbi matrix is defined as follows:

$$\begin{cases} V_i(t) = \max_{o_1, o_2, \ldots, o_{t-1}} P(S_1^{t-1} = s_1^{t-1}, S_t = i, O_1^t = o_1^t) \\ V_i(1) = P(S_1 = i, O_1 = o_1) = \pi_i \cdot p_{i1} \end{cases} \tag{26}$$

where $i = 1, 2, \ldots, m, t = 2, 3, \ldots, n$. The Viterbi formula is as follows:

$$\begin{cases} s_n = \arg\max_i V_i(n) \\ s_t = \arg\max_i (V_i(t) \cdot a_{i,s_{t+1}}) \end{cases} \tag{27}$$

where $t = n-1, \ldots, 2, 1$, and $s_n$ is the optimal hidden state when the observed state is $o_n$.

**4.2.2 Recognition results of driver intentions.** Several groups of the driver pedal signals are sampled, and the accelerator and brake pedal velocity signals are transformed into observed state sequences of the HMMs and input into the Viterbi algorithm program. After being 'decoded' by the Viterbi algorithm, the optimal hidden state sequences are output. Finally, the recognition of the driver intentions can be completed by matching the hidden states with the corresponding driver intentions.

The pedal signals were sampled using dSPACE and a simulator. During the process of signal acquisition, the driver operates the accelerator and brake pedal with different velocities to

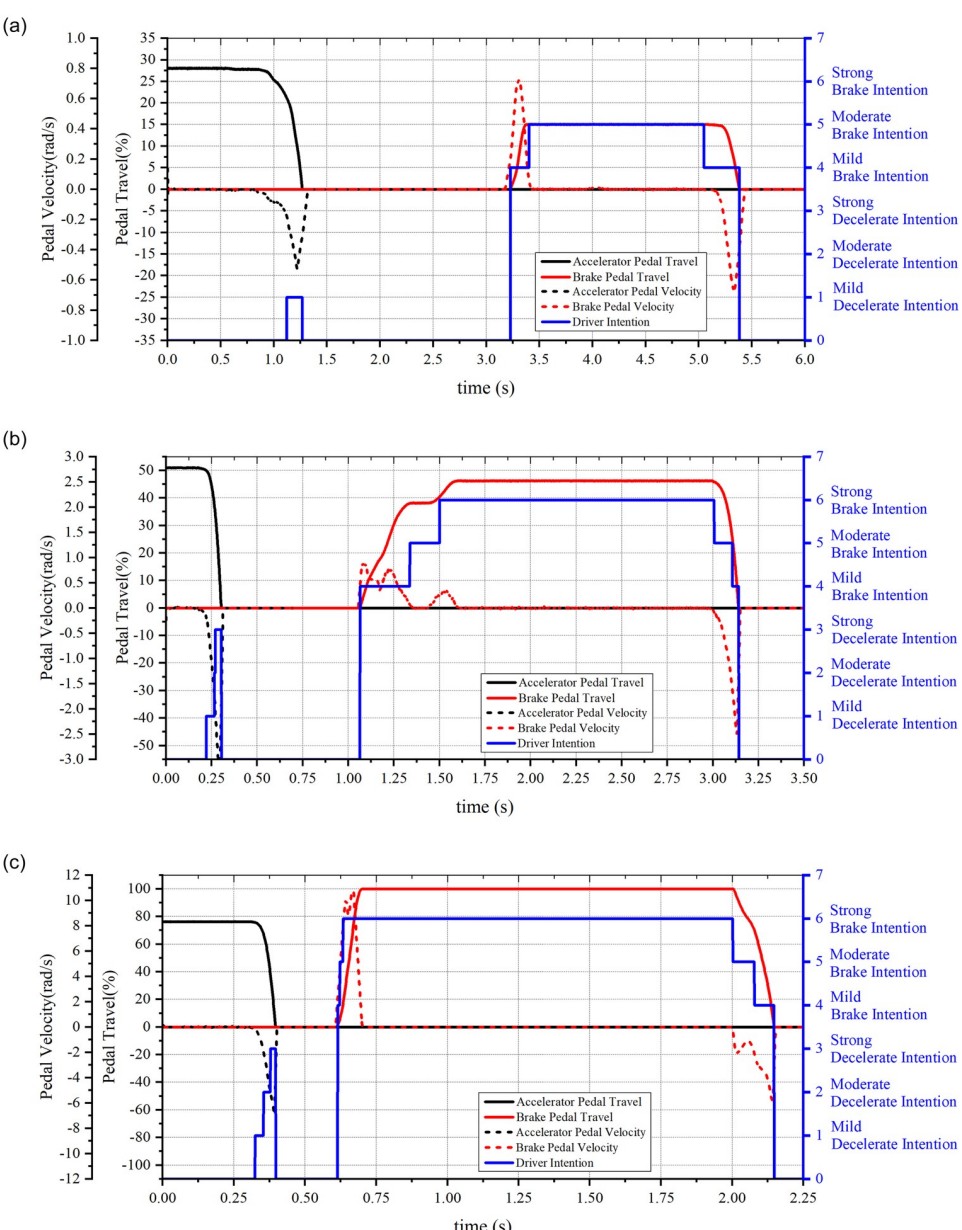

**Fig 11. Recognition results of driver intentions.** (a) Sample 1; (b) Sample 2; (c) Sample 3.

simulate the vehicle deceleration and braking condition under the driver control. The sampled pedal signals and recognized intentions are shown in Fig 11; the HMM hidden state values and the corresponding driver intentions are shown in the right axis of the figure.

It can be seen from the figure that the HMM based method can accurately recognize the driver intentions for different scenarios.

## 5. Clamping force control strategy based on driver intentions

Previous studies show that conventional clamping force control methods using cascade architectures often use clamping force as the sole control variable. In these methods, EMB

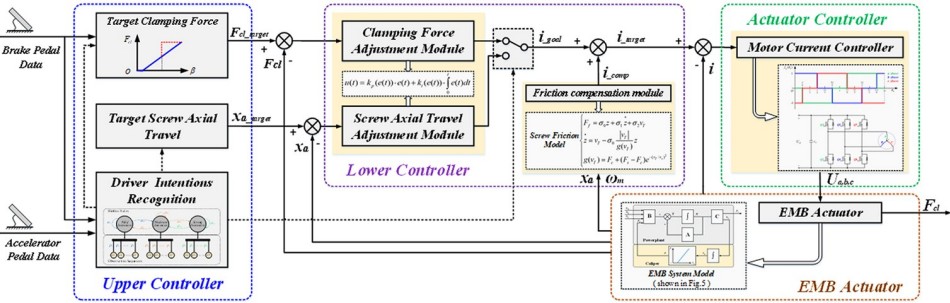

**Fig 12. Overall control architecture.**

eliminates the brake clearance and outputs the clamping force when the driver hits the brake pedal. However, the behavior of the proposed EMB (as described in section 1, paragraph 8) making these methods time–consuming, and thereby affecting the clamping force response performance for safe driving.

Studies show that the driver reaction time, from receiving the emergency brake signal to hitting the brake pedal, is approximately 0.3 s–1.0 s [27]. If driver intention can be predetermined, the driver reaction time can be effectively utilized to eliminate the brake clearance in advance and shorten the clamping force response time, thereby minimizing the adverse effects caused by the proposed EMB.

In this section, a clamping force control strategy based on the driver intentions is built, and the clamping force response and follow–up performances are simulated and analyzed under the typical signals. Further, the proposed control strategy is verified by performing simulations under the driver pedal signals.

## 5.1 Overall control architecture: Layered

In general, the clamping force control architecture adopts a layered structure composed of an upper, lower, and actuator controllers. First, the upper controller recognizes the driver's decelerate and brake intentions according to the established HMMs and determines the control variables and targets. The lower controller is composed of a clamping force and screw axial travel adjustment and friction compensation modules. The clamping force and screw axial travel adjustment modules are connected in parallel. The friction compensation module compensates for the nonlinear characteristics of friction in the screw mechanism. Finally, the actuator controller consists of a current hysteresis controller and an inverter, responsible for driving the BLDC motor. The overall clamping force control architecture is shown in Fig 12.

## 5.2 Upper controller: Determines the control variables and targets

Based on the recognition of the driver intentions and layered control architecture, this section comprehensively determines the control variables and targets of the EMB from the following aspects:

- When the driver has a strong decelerate intention:
  The upper controller considers the screw axial travel as the control variable, and the brake clearance is eliminated in advance by utilizing the reaction time needed to release the accelerator pedal and move the right foot to the brake pedal. Thus, when the driver hits the brake pedal, the EMB outputs the clamping force without delay.
  Based on this, the screw target axial travel is set to eliminate the brake clearance without

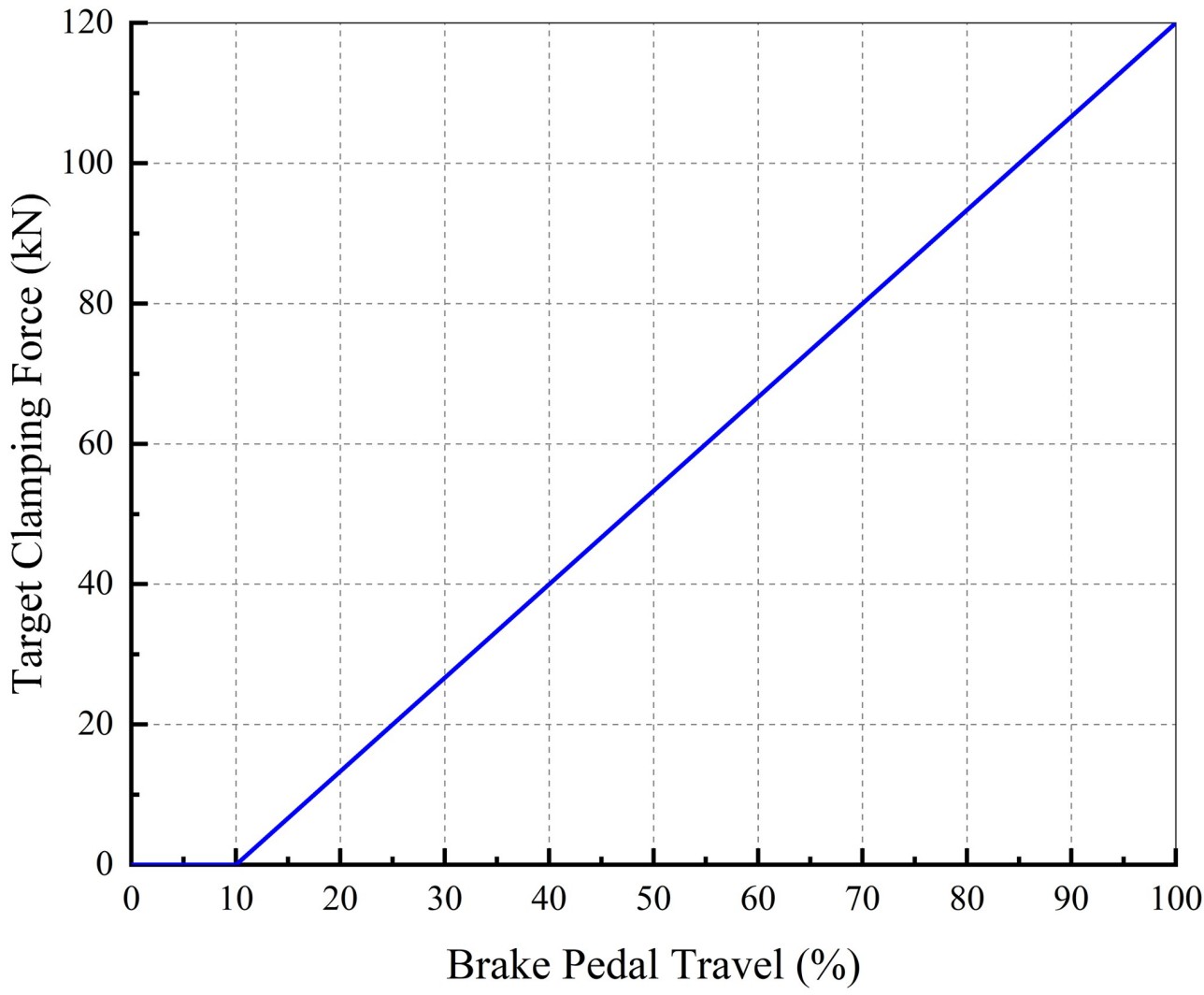

**Fig 13. Relationship between brake pedal travel and target clamping force.**

frequent contact and abnormal friction between the brake shoes and disc. Because of the self–adjusting mechanism of the clearance in the caliper, the braking clearance can be approximated to a certain value (approximately 0.5 mm one–side). Combined with the characteristic curve of the caliper (as shown in Fig 5), the screw target axial travel is set to 15 mm.

- When the driver hits the brake pedal:
  The upper controller considers the clamping force as the control variable and decides the target clamping force according to the curve shown in Fig 13. However, when the driver demonstrates a strong brake intention and the brake pedal travel exceeds 50%, the upper controller considers the maximum clamping force as the target immediately and does not decide the target clamping force based on Fig 13. This is called the clamping force enhanced strategy (CE).

- When the driver hits the accelerator pedal after braking:
  When the driver hits the accelerator pedal, it indicates that the driver has no intention to decelerate or brake in a short time. To ensure safe driving and avoid abnormal contact

between the brake shoes and disc, the upper controller exhibits screw axial travel as the control variable and the screw initial position as the target; screw reset is then required.

## 5.3 Lower controller: Realizes the adjustment of clamping force and screw axial travel, and compensates for the friction

**5.3.1 Adjustment of clamping force and screw axial travel.** The adjustments of the clamping force and screw axial travel are implemented using the nonlinear PI algorithm. The input of the clamping force adjustment module is the error between the target clamping force $F_{cl\_target}$ and actual force $F_{cl}$, and the output of this module is the goal current $i_{goal}$. For the screw axial travel adjustment module, the input is the error between the target value $x_{a\_target}$ and actual value $x_a$ of screw axial travel, and the output is the goal current $i_{goal}$.

Unlike the PI, the proportional and integral parameters in the nonlinear PI algorithm are functions of the control error $e(t)$, which are not constants, and vary with the error $e(t)$. Therefore, the mathematical expression of the nonlinear PI algorithm is:

$$u(t) = k_p(e(t)) \cdot e(t) + k_i(e(t)) \cdot \int_0^t e(t)dt \tag{28}$$

where,

$$k_p(e(t)) = a_p + b_p(1 - \frac{2}{e^{c_p e(t)} + e^{-c_p e(t)}}) \tag{29}$$

$$k_i(e(t)) = a_i \cdot \frac{2}{e^{c_i e(t)} + e^{-c_i e(t)}} \tag{30}$$

where $a_p$, $b_p$, $c_p$, $a_i$, $c_i$ are real numbers greater than zero. If they are appropriately selected, the control system can respond quickly without an overshoot.

**5.3.2 Compensation of friction.** The friction compensation is used to produce an additional torque by the motor to overcome the friction torque produced by the screw mechanism. This is realized by determining a compensation current $i_{comp}$. The calculation of $i_{comp}$ is combined with the friction model and characteristics of the BLDC motor. LuGre's friction model is used to describe the friction phenomenon of the screw mechanism [28], as given by (31):

$$\begin{cases} F_f = \sigma_0 z + \sigma_1 \cdot z + \sigma_2 v_T \\ \cdot z = v_T - \sigma_0 \dfrac{|v_T|}{g(v_T)} z \\ g(v_T) = F_c + (F_s - F_c)e^{-(v_T/v_s)^2} \end{cases} \tag{31}$$

where $F_f$ is the friction force; $\sigma_0$ is the rigidity coefficient; $\sigma_1$ and $\sigma_2$ are the viscous damping and friction coefficients, respectively; $g(v_T)$ is the Stribeck function; $F_c$ and $F_s$ are the Coulomb and static frictions, respectively; $v_s$ is the Stribeck velocity. Because of the surface contact between the screw and head, the relative velocity $v_T$ of the screw and head along the inclined surface is given by:

$$v_T = \omega_m \frac{d_m}{2} \cdot \cos(\alpha + \rho) - \cdot x_a \sin(\alpha + \rho) \tag{32}$$

In addition, when the motor operates in $2\pi/3$ conduction mode, and the transient process of commutation is ignored, then the characteristics of the motor can be simplified as [29]:

$$T_e = K_T i \tag{33}$$

where $i$ is the motor current at steady state. Thus, the current required to compensate the friction is:

$$i_{comp} = \frac{T_f}{K_T} = \frac{F_f \cdot d_m}{2K_T} \tag{34}$$

## 5.4 Actuator controller: Drives the BLDC motor

The actuator controller consists of a current hysteresis controller and an inverter, which drives the BLDC motor through a square wave. The inverter adopts a three–phase bridge and two-way conduction mode [30]. The principle of the current hysteresis controller is that when the difference between the reference and actual currents reaches the positive edge of the hysteresis width, a pulse is generated to turn on the corresponding power switching component of the inverter; else, the corresponding power switching unit is turned off. Here, the actual current is measured by the sensor and reference current $i_{R\_abc}$ is calculated as:

$$i_{R\_abc} = i_{target} \times f_{iR}(\theta_m) \tag{35}$$

where $f_{iR}(\theta_m)$ is the reference current waveform as shown in Fig 14, which corresponds to the back EMF waveform of the motor (shown in Fig 3). In addition, the total target current $i_{target}$ is output from the lower controller and consists of $i_{goal}$ and $i_{comp}$.

## 5.5 Simulation analyses

**5.5.1 Clamping force response and follow-up performance of the EMB.**   To analyze the response and follow-up performances, the target clamping forces of the step and triangular sawtooth signals were input into the EMB and simulated using MATLAB/Simulink. The parameters of the control algorithm are listed in Table 6 and the corresponding simulation results are shown in Figs 15 and 16.

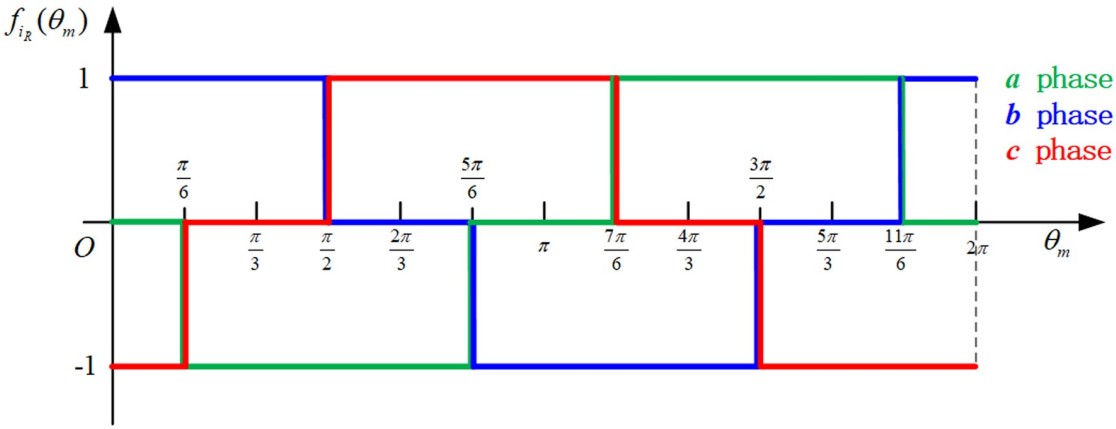

**Fig 14. Reference current waveform.**

**Table 6. Parameters of control algorithm.**

| PI | | Nonlinear PI | | | |
|---|---|---|---|---|---|
| $k_p$ | 1500 | $a_p$ | 1500 | $a_i$ | 0.15 |
| $k_t$ | 0.15 | $b_p$ | 0.5 | $c_i$ | 0.01 |
| | | $c_p$ | 0.5 | | |

The maximum clamping force of the step signal input into the EMB and the corresponding simulation results are shown in Fig 15.

As observed from Fig 15a, compared with the PI control algorithm, the clamping force response curve based on nonlinear PI algorithm changes smoothly, without any overshoot or jitter, when approaching the target. The EMB eliminates the brake clearance at approximately 0.2 s, and the clamping force reaches the maximum value of 120 kN at 0.75 s. From Fig 15b,

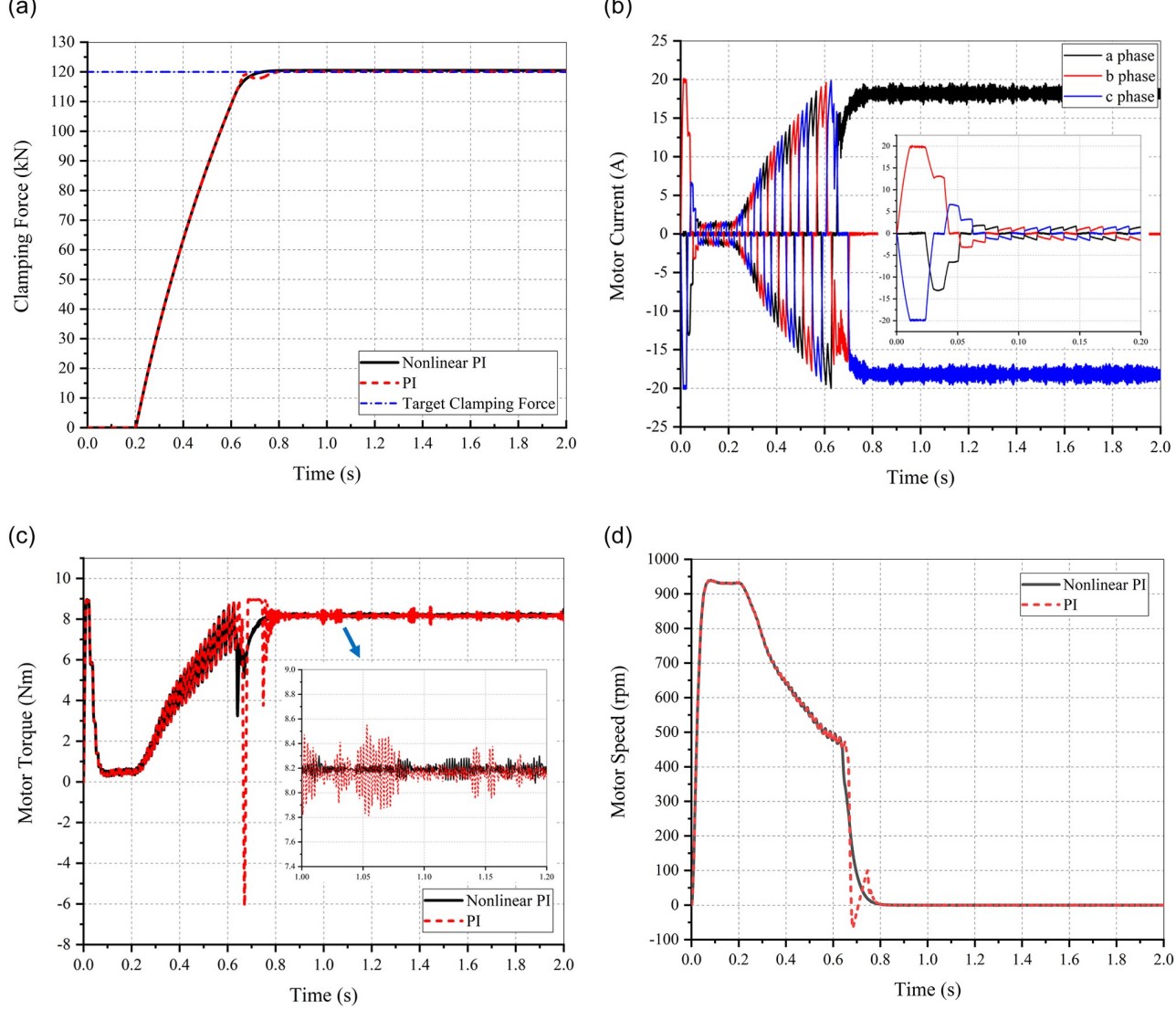

**Fig 15. Simulation results of the clamping force response performance.** (a) Clamping force. (b) Motor current. (c) Motor torque. (d) Motor speed.

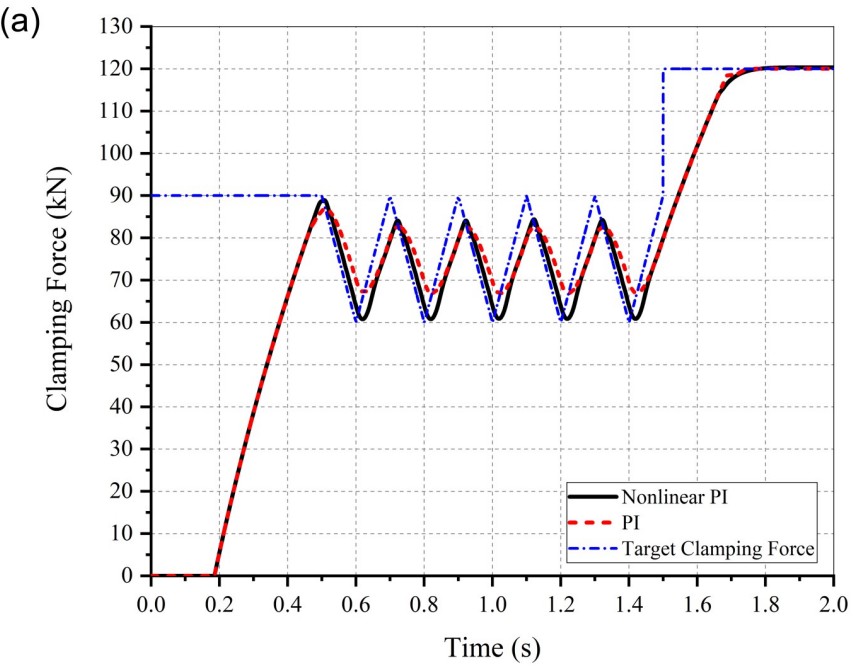

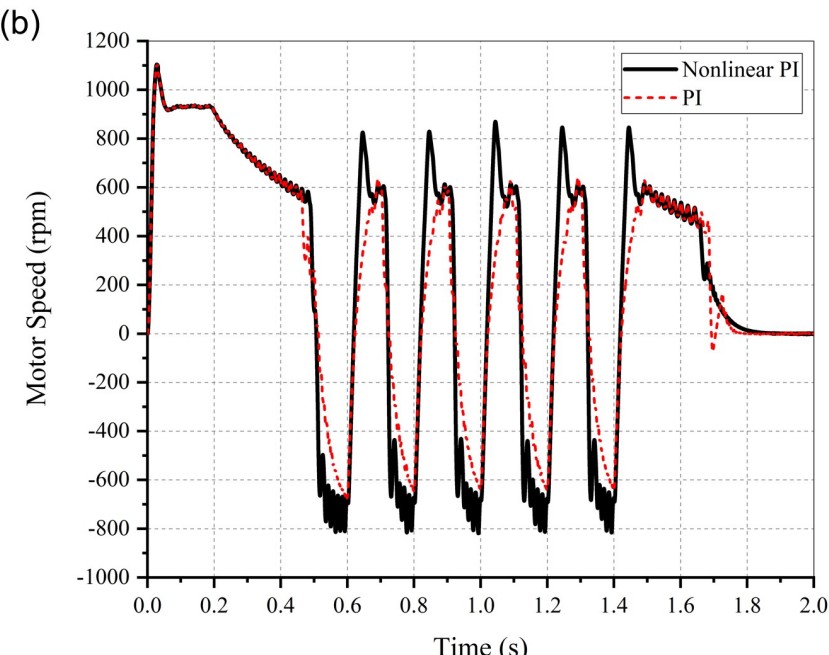

**Fig 16. Simulation results of the clamping force follow–up performance.** (a) Clamping force. (b) Motor speed.

the motor current trends are seen to be consistent with the reference current waveform shown in Fig 14. It can be concluded from Fig 15c and 15d that the nonlinear PI algorithm can attenuate the fluctuations in motor torque and speed when approaching or reaching the target.

The target clamping force of the triangular sawtooth signals were input into the system to simulate changes in clamping force when the ABS is triggered. The target signal fluctuates between 60 kN and 90 kN at 5 Hz. The corresponding simulation results are shown in Fig 16.

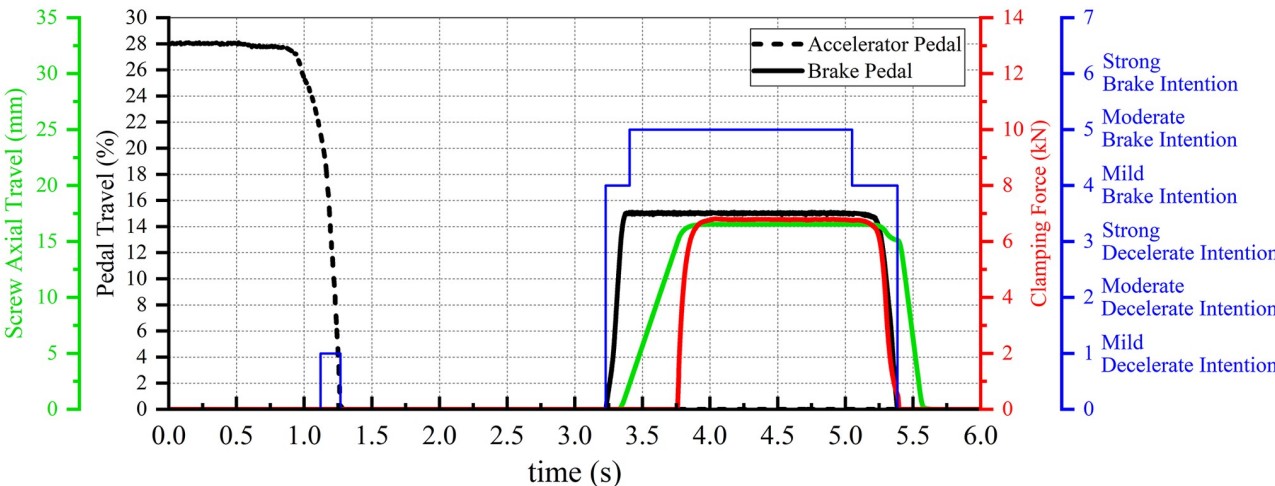

**Fig 17. Simulation result 1 of the clamping control strategy.**

It can be observed that the switch between the forward and reverse rotation of the motor is faster and more flexible when the nonlinear PI control algorithm is adopted under the action of the aforementioned signal. The follow–up performance of the clamping force is also better than the PI algorithm.

**5.5.2 Clamping force control strategy considering driver's intentions.** The pedal signals shown in Fig 11 are used to simulate and analyze the proposed control strategy in MATLAB/ Simulink. The pedal signals, depicted in Fig 11a, form the initial input; the simulation results are shown in Fig 17.

From the figure, it can be seen that to avoid frequent braking actions, the EMB does not eliminate brake clearance in advance under the driver mild decelerate intention. When the driver hits the brake pedal and the brake clearance is eliminated, the clamping force becomes the output.

Further, the pedal signals, illustrated in Fig 11b, are input into the strategy, and the corresponding simulation results are shown in Fig 18.

It can be seen from the figure that the driver starts to release accelerator pedal at A. Subsequently, when the upper controller recognizes that the driver has a strong decelerate intention, the EMB begins to eliminate the brake clearance in advance. In this process, the screw extends

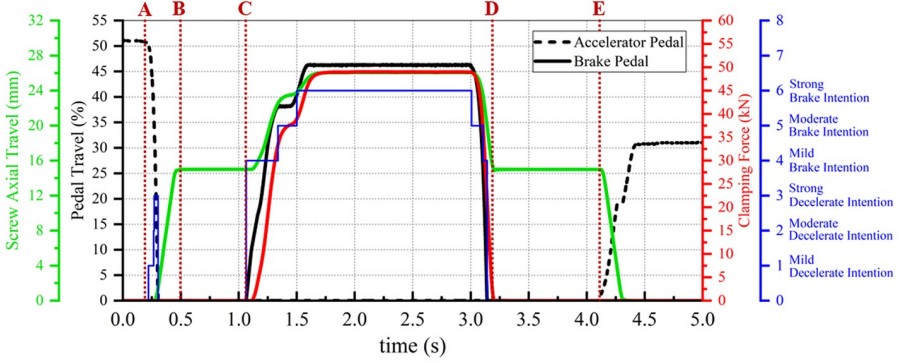

**Fig 18. Simulation result 2 of the clamping control strategy.**

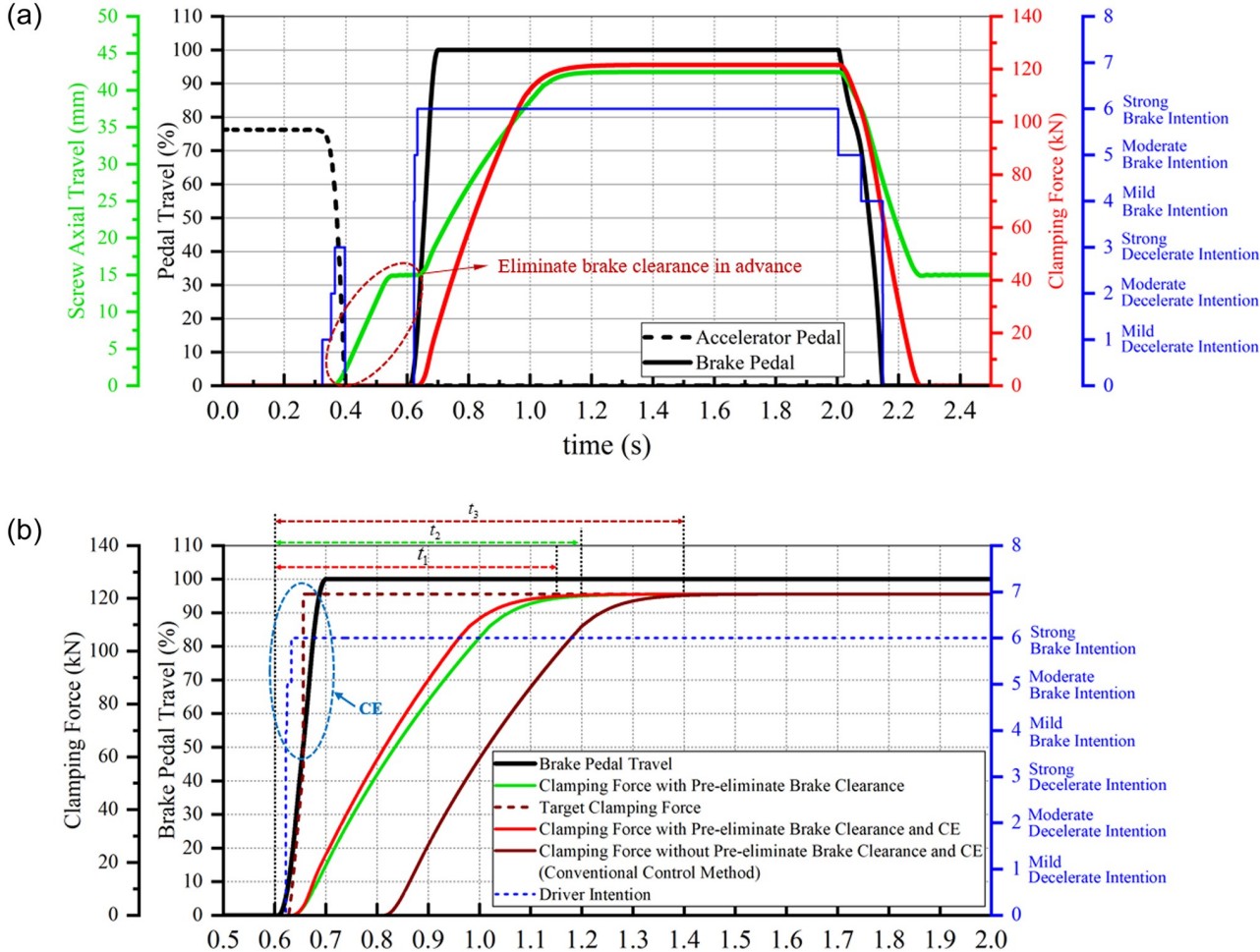

**Fig 19. Simulation result 3 of the clamping control strategy.** (a) Clamping force control. (b) Comparison of control effects: with and without pre-elimination brake clearance, and with and without CE.

axially, under the drive of the motor, to B; the screw is always at the waiting position from B to C. As the driver hits the brake pedal at C, the system outputs the clamping force until the driver completely releases the brake pedal at D. As the clamping force is removed, the screw returns to the waiting position. Finally, when the driver hits the accelerator pedal, the screw returns to the initial position to avoid unnecessary braking caused by the abnormal contact between the brake shoes and disc.

The pedal signals, depicted in Fig 11c, represent the emergency brake operation input into the strategy. The corresponding simulation results are shown in Fig 19. Fig 19a shows the driver intentions, pedal signals and EMB operations. It is observed that when EMB recognizes that the driver has a strong decelerate intention, it eliminates the brake clearance before the driver hits the brake pedal. Thus, when the driver forcefully hits the brake pedal, the clamping force is output on time.

Fig 19b shows the comparison of control effects under the driver strong brake intention, where the simulation results of 0.5 s–2.0 s are intercepted. As observed from the figure, owing to the pre–elimination of brake clearance and CE, the EMB outputs the clamping force almost

**Table 7. Simulation results.**

| Input signals | Clamping force response time | | |
|---|---|---|---|
| Step signal of maximum clamping force | PI control algorithm and conventional control method | nonlinear PI control algorithm and conventional control method | |
| | 0.8 s | 0.75 s | |
| Driver pedal signals | Without pre–elimination of the brake clearance and CE | With pre–elimination of the brake clearance, but without CE | With pre–elimination of the brake clearance and CE |
| | $t_3 = 0.8$ s | $t_2 = 0.6$ s | $t_1 = 0.55$ s |

without a delay when the driver hits the brake pedal, and quickly attains the maximum clamping force as the control target when the brake pedal travels more than 50% (shown with the brown dotted line in Fig 19b).

Under the combined actions (shown with the red solid line in Fig 19b), the clamping force response time is shortened by approximately 0.25 s compared with the conventional method (based on cascade architecture using clamping force as the sole control variable, as shown with the brown solid line in Fig 19b), which improves the EMB response performance. The negative effect, attributed to the leverage principle of the proposed EMB, on the response performance is almost eliminated. The corresponding simulation results are listed in Table 7.

## 6. Conclusion

Contrary to other studies, the proposed EMB based on pneumatic disc brake demonstrates a longer clamping force response time under the conventional control method, which is not ideal for safe driving. Based on this observation, the working principle of the proposed EMB was analyzed, and its system model was established. In addition, a clamping force control strategy was designed based on driver intentions. Notably, the following work was conducted:

First, the relationship between vehicle driving conditions, driver intentions, and pedal behaviors were analyzed. The HMMs of driver's decelerate and brake intentions were built, and the respective parameters were trained.

Next, a clamping force control strategy based on driver intentions was proposed to minimize the scheme's adverse effects without changing the structure and size of the EMB. Particularly, the brake clearance was eliminated in advance according to the driver's decelerate intention, and the target clamping force was adjusted on time according to the driver's brake intention to improve the overall clamping force response performance.

Finally, simulation analyses are performed using MATLAB/Simulink. The results show that:

- When using nonlinear PI algorithms, the clamping force response curve gradually increases, without any overshoot or jitter, when approaching at target step signal. It takes approximately 0.2 s to eliminate the brake clearance, while the maximum clamping force of 120 kN is attained at 0.75 s.

- When the target clamping force is a triangular sawtooth signal of 5 Hz, EMB can identify the target signal. The switch between the forward and reverse rotation of the motor is fast and flexible based on the nonlinear PI algorithm.

- The established HMMs can accurately recognize the driver's intentions, while the proposed control strategy enables EMB to achieve brake clearance pre–elimination and restoration at appropriate times.

- By combining driver intention recognition, brake clearance pre–elimination, and CE, the overall clamping force response time was shortened by approximately 0.25 s under the driver emergency brake compared with the conventional control method.

Future work can address fabricating a prototype of the EMB for field and road tests to validate the proposed control strategy further.

## Supporting information

**S1 Appendix. Judgement matrices.**
(DOCX)

**S1 File. Simulation results.**
(ZIP)

## Acknowledgments

We would like to thank *Editage* for English language editing.

## Author Contributions

**Data curation:** Yan He, Lingshuai Meng.

**Formal analysis:** Tong Wu, Zuoyue Han.

**Funding acquisition:** Jing Li.

**Investigation:** Tianxin Fan.

**Methodology:** Tong Wu.

**Project administration:** Jing Li.

**Software:** Tianxin Fan.

**Validation:** Lingshuai Meng.

**Writing – original draft:** Tong Wu, Yan He.

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
