## [Decision Letter · Decision Letter 0]

29 Apr 2020

PONE-D-20-05557

Clamping force control of electro-mechanical brake ： Based on driver's intentions

PLOS ONE

Dear Tong Wu,

Thank you for submitting your manuscript to PLOS ONE. After careful consideration, we feel that it has merit but does not fully meet PLOS ONE’s publication criteria as it currently stands. Therefore, we invite you to submit a revised version of the manuscript that addresses the points raised during the review process.

We would appreciate receiving your revised manuscript. To enhance the reproducibility of your results, we recommend that if applicable you deposit your laboratory protocols in protocols.io, where a protocol can be assigned its own identifier (DOI) such that it can be cited independently in the future. For instructions see: http://journals.plos.org/plosone/s/submission-guidelines#loc-laboratory-protocols

We look forward to receiving your revised manuscript.

Kind regards,

Chen Lv, PhD

Academic Editor

PLOS ONE

Journal Requirements:

"JL received support from the National Key R&D Program of China (2018YFB0105900).

The funders had no role in study design, data collection and analysis, decision to

publish, or preparation of the manuscript." 

We note that one or more of the authors are employed by a commercial company: "UISEE(Shanghai) Automotive Technologies Ltd,"

Additional Editor Comments (if provided):

Based on reviewers' comments, a major revision is required. Please carefully address all the points raised by reviewers. Besides, since there are authors being affiliated to UISEE(Shanghai) Automotive Technologies Ltd, based on journal's policy, please declare if there is any potential competing interest.

Reviewers' comments:

Reviewer's Responses to Questions

**Comments to the Author**

1. Is the manuscript technically sound, and do the data support the conclusions?

Reviewer #1: Yes

Reviewer #2: Yes

2. Has the statistical analysis been performed appropriately and rigorously? 

Reviewer #1: Yes

Reviewer #2: Yes

3. Have the authors made all data underlying the findings in their manuscript fully available?

Reviewer #1: Yes

Reviewer #2: Yes

4. Is the manuscript presented in an intelligible fashion and written in standard English?

Reviewer #1: Yes

Reviewer #2: Yes

5. Review Comments to the Author

Reviewer #1: This paper gives a good research on clamping force control of EMB system. However, I still have some questions listed as follows.

1. As we all know, EMB has not widely used in vehicle bracking system becaused of its space layout. Now, EHB is an good solution of

bracking system for autonomous vehicle. The firgue of EMB in this paper is desigend by the authors? If yes, have you made prototype?

It is necessary to give the advantage of the proposed EMB, compared with other EMBs and EHB system.

2. Since the EMB system is a complex mechanical system, can the proposed model for algorithm evaluation simulate the real mechanical

system? You should give description. If a prototype is made, HIL test may be more effective.

3. Why do you choose nonlinear PI controller rather than SMC or other control algorithms? SMC is a very good method to deal with nonlinear

system.

Reviewer #2: There are large amount of articles focusing on the development of EMB and the initial idea of your paper is quite like the development base of BBW (brake by wire) system. Hence, the innovation point should be highlighted in comparison with the current BBW system. Otherwise, the comparison with BBW should be clarified carefully.

6. PLOS authors have the option to publish the peer review history of their article (what does this mean?). If published, this will include your full peer review and any attached files.

Reviewer #1: No

Reviewer #2: No

---

## [Author Response · Author response to Decision Letter 0]

15 Jul 2020

Dear Editors and Reviewers:

 Thank you for your letter and for the reviewers’ comments concerning our manuscript entitled “Clamping force control of electro-mechanical brakes based on driver intentions” (Manuscript Number: PONE-D-20-05557 and PONE-D-20-05557R1). Those comments are all valuable and very helpful for revising and improving our paper, as well as the important guiding significance to our researches.

 First of all, most text of the previous manuscript has been copyedited. These changes will not influence the content and framework of the paper, and highlighted by blue in the updated manuscript.

 Furthermore, we have studied comments carefully and have made correction which we hope meet with approval. Revised portion are highlighted by red in the updated manuscript. The main corrections in the paper and the responds to the reviewer’s comments are as flowing:

Reviewer #1: This paper gives a good research on clamping force control of EMB system. However, I still have some questions listed as follows.

1. As we all know, EMB has not widely used in vehicle braking system because of its space layout. Now, EHB is an good solution of braking system for autonomous vehicle. The figure of EMB in this paper is designed by the authors? If yes, have you made prototype?

It is necessary to give the advantage of the proposed EMB, compared with other EMBs and EHB system.

Response: We greatly appreciate your encouraging comments and thank you for your insightful suggestions. For question 1, our reply is as follows:

(1) The EMB proposed in this paper is designed by ourselves, and the prototype is being fabricated and matched. Next, we will perform HIL test and road test.

(2) We revised Section 1 to reflect the advantages of the proposed EMB compared with other EMBs and EHB systems:

 1) First, we revised the 3rd paragraph of Section 1 (line 46~51 and line 54~59) to summarize and compare the structure of current EMBs, and added Fig 1 at the end of this paragraph:

“Motors are used as the power source in most schemes, while a screw or bevel gear [6, 11] is used for the motion conversion mechanism. Particularly, the existing EMBs can be divided into two conventional schemes based on the force amplifying mechanism as shown in Fig 1. In scheme A, as shown in Fig 1a, a gear [4–7] or worm [8, 9] is used as the force amplifying mechanism; scheme B, shown in Fig 1b, utilizes the self–increasing effect of wedge mechanisms to increase clamping force [9–12].”

And“Contrarily, scheme A is more direct in power transmission and simple in control, but the output capacity of the system is dependent on the motor and reduction ratio, which requires a higher performance of the motor. Although scheme B utilizes the wedge mechanism to increase the force, which demonstrates a flexible arrangement and lower requirements of the motor, the strong nonlinear characteristics of the wedge mechanism increases the control difficulty and decreases the braking stability [12].”

 2) Undeniably, several enterprises, institutions and scholars have been investigating on EHB, they proposed many schemes and performed road tests, and even equipped EHB in some mass-produced cars. While, limited by storing and sealing of hydraulic oil, EHB hardly to apply in heavy commercial vehicles. Thus, we revised the 7th paragraph of Section 1 (line 89~91):

“Constraints in storing and sealing of hydraulic oil make it difficult to employ EHB in commercial vehicles, particularly in heavy commercial vehicles. Currently, most commercial vehicles are equipped with pneumatic brake systems.”

 3) Furthermore, we revised the 8th paragraph of Section 1 (line 104~112), and compared the proposed EMB scheme with existing EMBs to highlight the advantages of the scheme:

“Compared with the existing EMB schemes, the proposed EMB is driven by the motor directly, and caliper is equipped with an automatic clearance adjustment device; hence, it has the advantages of direct transmission and simple control as that of scheme A (shown in Fig 1a). In addition, the arm inside the caliper can enlarge the axial thrust force output from the powerplant; thus, the output capacity of the EMB is improved, and the performance requirements of the motor are reduced. The proposed EMB satisfies the brake demand of commercial vehicles and overcomes the control difficulty and high nonlinearity of scheme B (shown in Fig 1b). Furthermore, the pneumatic brake caliper is retained in the proposed EMB, making it simple, flexible, and versatile; this scheme can be conveniently transferred from pneumatic brake to EMB.”

2. Since the EMB system is a complex mechanical system, can the proposed model for algorithm evaluation simulate the real mechanical system? You should give description. If a prototype is made, HIL test may be more effective.

Response: For question 2, our reply is as follows:

(1) Your question concerns about the accuracy of the established EMB model, we are fully aware that it is necessary. Therefore, we divide the EMB model into two parts: the powerplant and the caliper. For the powerplant, we deduced its model according to electrical fundaments and kinematics based on the designed parameters (listed in Table 1) and references [7], [15] and [16]. Furthermore, owing to the complexity of the caliper, we use the characteristic of the pneumatic disc brake as the caliper model to ensure the accuracy. The original data of the characteristic is provided by the caliper manufacturer, which can simulate the real mechanical system;

(2) Hence, we revised Section 2 as follows:

 1) Add a paragraph (line 151~156) under the title "EMB System modeling" in Section 2.2 to introduce the general idea of model establishment:

“The EMB is a complex mechanical system, and the accuracy of its model affects the control performance. Therefore, according to the scheme, the EMB system model is divided into two parts: powerplant, and caliper. The powerplant model is composed of a BLDC motor and screw mechanism, fabricated according to the electrical fundamentals and kinematics. The caliper model is complex and described using manufacturer specifications and fitted in this study to ensure accuracy and reduce complexity.”

 2) Add a paragraph (line 158~159) under the title "Crucial part: Powerplant" in Section 2.2.1 to describe the function of the powerplant:

“The powerplant axially extends the screw under the BLDC motor drive, and convert the motor torque into an axial thrust force.”

 3) Revised Section 2.2.2 (line 201~214) to describe the functions, composition and establishment ideas of the caliper model:

“There are two main functions of the caliper:

 1) To amplify the axial thrust force output from the powerplant and transform it into the clamping force through the action between the brake shoes and disc;

 2) Simultaneously, the caliper applies the reaction force to the powerplant, according to Newton's third law, for the EMB system to attain a balanced state. At this point, the caliper is equivalent to a "load device".

Therefore, the caliper model should accurately reflects the "load". In this study, the characteristic of a pneumatic disc brake is used for the caliper model, which describes the relationship between and . Here, is the axial travel of chamber push rod and is the force between brake shoes and disc. According to the pneumatic brake working principle, corresponds to the screw axial travel distance xa in the proposed EMB, and is equal to the load force FL of the caliper acting on the powerplant. Fig 5 shows the characteristic of a pneumatic disc brake; a series of discrete points in the figure represent the measured data provided by the caliper manufacturer, and the blue dotted line is the fitted characteristic curve. The caliper model is further observed in the figure.”

(3) Prototypes are being processed and matched. Next, we will testify the accuracy of the model using the prototype.

3. Why do you choose nonlinear PI controller rather than SMC or other control algorithms? SMC is a very good method to deal with nonlinear system.

Response: For question 3, our reply is as follows:

(1) Undoubtedly, SMC has many advantages, such as robustness, which is a good method to deal with nonlinear system. However, the chattering of SMC has not been well solved, which also reflected in the literature [18] etc. In this study, if the clamping force chatters during the braking process, an additional longitudinal impact will be generated, which may affect the braking quality even safe driving. Therefore, during the braking process, the clamping force is required to change smoothly, and there is without an overshoot and jitter as much as possible;

(2) Unlike the PI, parameters in the nonlinear PI controller can vary with the error nonlinearly, which are not constants. This indicates that the system which uses nonlinear PI algorithm can respond quickly without an overshoot if the parameters appropriately selected. (mentioned in line 488~493);

(3) Furthermore, the nonlinear PI controller does not depend on the controlled plant model, and the parameter tuning is relatively easy. These make the proposed EMB and its clamping force control strategy simple, flexible and versatile. Combining (1) ~ (3), these are the main reasons why we choose nonlinear PI algorithm;

(4) Thank you for your insightful suggestion. We think it's very meaningful to combine SMC with other intelligent algorithms or optimization theories to conduct in-depth research on its chattering problem.

Reviewer #2: There are large amount of articles focusing on the development of EMB and the initial idea of your paper is quite like the development base of BBW (brake by wire) system. Hence, the innovation point should be highlighted in comparison with the current BBW system. Otherwise, the comparison with BBW should be clarified carefully.

Response: We greatly appreciate your encouraging comments and valuable suggestions. For your suggestion, our reply is as follows:

(1) Undoubtedly, we have to acknowledge that there are many articles about EMB, which give us a meaningful reference in our research. So, we have listed some representative articles as references;

(2) Owing to the behavior of the proposed EMB (8th paragraph of section 1, line 112~117), it is time–consuming for the EMB to eliminate the brake clearance, which affects the clamping force response performance for safe driving. Thus, the original idea of this study is to improve the clamping fore response time without changing the structure and size of the EMB. The original text is as follows:

“However, we observed a problem during the process of research: although the force arm inside the caliper can magnify the axial thrust force output from the powerplant and act on the brake shoes, it enlarges the axial displacement of the brake shoes into the screw, according to the leverage principle (shown in Fig 2a). Limited by the ultimate performance of the motor, it is time–consuming to eliminate brake clearance, which affects the EMB performance and driving safety, which is unfavorable.”（8th paragraph of Section 1，line 112~117）

And “To minimize the scheme’s adverse effects, the primary focus of this study is to improve the clamping force response performance without changing the structure and size of the EMB.”（9th paragraph of Section 1, line 122~123）

(3) As the reviewer’s request, we revised Section 1 to reflect the innovation of the proposed EMB compared with the current BBW system:

 1) First, we revised the 3rd paragraph of Section 1 (line 46~51 and line 54~59) to summarize and compare the structure of current EMBs, and add Fig 1 at the end of the paragraph for explanation:

“Motors are used as the power source in most schemes, while a screw or bevel gear [6, 11] is used for the motion conversion mechanism. Particularly, the existing EMBs can be divided into two conventional schemes based on the force amplifying mechanism as shown in Fig 1. In scheme A, as shown in Fig 1a, a gear [4–7] or worm [8, 9] is used as the force amplifying mechanism; scheme B, shown in Fig 1b, utilizes the self–increasing effect of wedge mechanisms to increase clamping force [9–12].”

And“Contrarily, scheme A is more direct in power transmission and simple in control, but the output capacity of the system is dependent on the motor and reduction ratio, which requires a higher performance of the motor. Although scheme B utilizes the wedge mechanism to increase the force, which demonstrates a flexible arrangement and lower requirements of the motor, the strong nonlinear characteristics of the wedge mechanism increases the control difficulty and decreases the braking stability [12].”

 2) Second, we revised 7th paragraph of Section 1 (line 89~91 and line 94~95) to explained the reason why EHB is difficult to be applied in commercial vehicles, and the brilliant prospect of EMB in commercial vehicles is pointed out by comparing it with pneumatic brake system:

“Constraints in storing and sealing of hydraulic oil make it difficult to employ EHB in commercial vehicles, particularly in heavy commercial vehicles. Currently, most commercial vehicles are equipped with pneumatic brake systems. Compared with the pneumatic brake, the EMB does not require large and complicated pneumatic pipelines and components. Furthermore, the EMB could effectively reduce the body weight and complexity of the braking system, eliminate the exhaust noise during braking, and realize more accurate control and distribution of the braking force. Therefore, the EMB demonstrates broader prospects than the pneumatic brake in the applications of commercial vehicles.”

 3) Furthermore, we revised the 8th paragraph of Section 1 (line 104~112) and compared the proposed EMB with current EMBs to highlight the innovation of the proposed scheme:

“Compared with the existing EMB schemes, the proposed EMB is driven by the motor directly, and caliper is equipped with an automatic clearance adjustment device; hence, it has the advantages of direct transmission and simple control as that of scheme A (shown in Fig 1a). In addition, the arm inside the caliper can enlarge the axial thrust force output from the powerplant; thus, the output capacity of the EMB is improved, and the performance requirements of the motor are reduced. The proposed EMB satisfies the brake demand of commercial vehicles and overcomes the control difficulty and high nonlinearity of scheme B (shown in Fig 1b). Furthermore, the pneumatic brake caliper is retained in the proposed EMB, making it simple, flexible, and versatile; this scheme can be conveniently transferred from pneumatic brake to EMB.”

 We tried our best to improve the manuscript and made some changes in the manuscript. These changes will not influence the content and framework of the paper.

 We appreciate for Editors/Reviewers’ warm work earnestly, and hope that the correction will meet with approval.

 Once again, thank you very much for your comments and suggestions.

---

## [Decision Letter · Decision Letter 1]

10 Sep 2020

Clamping force control of electro-mechanical brakes based on driver intentions

PONE-D-20-05557R1

Dear Dr. Tong Wu,

We’re pleased to inform you that your manuscript has been judged scientifically suitable for publication and will be formally accepted for publication once it meets all outstanding technical requirements.

Kind regards,

Chen Lv, PhD

Academic Editor

PLOS ONE

Additional Editor Comments (optional):

Reviewers' comments:

Reviewer's Responses to Questions

**Comments to the Author**

1. If the authors have adequately addressed your comments raised in a previous round of review and you feel that this manuscript is now acceptable for publication, you may indicate that here to bypass the “Comments to the Author” section, enter your conflict of interest statement in the “Confidential to Editor” section, and submit your "Accept" recommendation.

Reviewer #1: All comments have been addressed

2. Is the manuscript technically sound, and do the data support the conclusions?

Reviewer #1: Yes

3. Has the statistical analysis been performed appropriately and rigorously? 

Reviewer #1: Yes

4. Have the authors made all data underlying the findings in their manuscript fully available?

Reviewer #1: Yes

5. Is the manuscript presented in an intelligible fashion and written in standard English?

Reviewer #1: Yes

6. Review Comments to the Author

Reviewer #1: (No Response)

7. PLOS authors have the option to publish the peer review history of their article (what does this mean?). If published, this will include your full peer review and any attached files.

Reviewer #1: No

---

## [Editor Report · Acceptance letter]

14 Sep 2020

PONE-D-20-05557R1

Clamping force control of electro–mechanical brakes based on driver intentions

Dear Dr. Wu:

I'm pleased to inform you that your manuscript has been deemed suitable for publication in PLOS ONE. Congratulations! Your manuscript is now with our production department.

Kind regards,

on behalf of

Dr. Chen Lv 

Academic Editor

PLOS ONE